# Neural-like computing with populations of superparamagnetic basis functions

Alice Mizrahi[1,2,4], Tifenn Hirtzlin[2], Akio Fukushima[3], Hitoshi Kubota [3], Shinji Yuasa[3], Julie Grollier[1] & Damien Querlioz[2]

In neuroscience, population coding theory demonstrates that neural assemblies can achieve fault-tolerant information processing. Mapped to nanoelectronics, this strategy could allow for reliable computing with scaled-down, noisy, imperfect devices. Doing so requires that the population components form a set of basis functions in terms of their response functions to inputs, offering a physical substrate for computing. Such a population can be implemented with CMOS technology, but the corresponding circuits have high area or energy requirements. Here, we show that nanoscale magnetic tunnel junctions can instead be assembled to meet these requirements. We demonstrate experimentally that a population of nine junctions can implement a basis set of functions, providing the data to achieve, for example, the generation of cursive letters. We design hybrid magnetic-CMOS systems based on interlinked populations of junctions and show that they can learn to realize non-linear variability-resilient transformations with a low imprint area and low power.

[1] Unité Mixte de Physique CNRS, Thales, Univ. Paris-Sud, Université Paris-Saclay, 91767 Palaiseau, France. [2] Centre de Nanosciences et de Nanotechnologies, Univ. Paris-Sud, CNRS, Université Paris-Saclay, 91405 Orsay, France. [3] Spintronics Research Center, National Institute of Advanced Industrial Science and Technology (AIST), Tsukuba, Ibaraki 305-8568, Japan. [4] Present address: Center for Nanoscale Science and Technology, National Institute of Standards and Technology, Gaithersburg, MD 20899-6202, USA. Correspondence and requests for materials should be addressed to J.G. (email: julie.grollier@cnrs-thales.fr)

The challenges to reduce the area and increase the energy efficiency of microelectronic circuits are increasing dramatically. The size of transistors is reaching the nanoscale, and decreasing their dimensions further, or using emerging nanometer-scale devices, leads to stochastic behaviors, large device-to-device variability, and failures[1,2]. Our current computing schemes are not able to deal well with noisy, variable, and faulty components. Entire processor chips are rejected based on a single component failure. However, we know that other forms of information processing can be extremely resilient to errors. Operating at the thermal limit, our brain seems to have found an optimal tradeoff between low-energy consumption and computational reliability[3]. It carries out amazingly complex computations even though its components, neurons, are very noisy[4,5]. Figure 1b illustrates a neural firing pattern triggered by a constant input stimulus: the periodicity of the spike train is typically blurred by the high level of noise.

A key reason for the resilience of the brain seems to be redundancy. Measurements of neuronal activity in diverse parts of the brain such as the retina[6], the midbrain[7], the motor cortex[8] or the visual cortex[9] indicate that these parts encode and process information by populations of neurons rather than by single neurons. This principle of population coding and its benefits for the brain have been investigated in numerous theoretical works[10,11]. In electronics, mimicking population coding has been proposed and shown to be effective in circuits using conventional

transistors, but leads to circuits with high area costs due to the large size of the artificial neurons[12,13]. It is therefore attractive to take inspiration from this strategy and compute with populations of low-area nanoscale electronic devices, even when they exhibit stochastic or variable behaviors. This approach has recently inspired pioneering studies of the dynamical response of ensembles of emerging nanodevices[14,15]. However, showing that actual computations can be realized using the physics of population of nanodevices remains an open challenge.

Neuroscience studies indicate that, for this purpose, elementary devices mimicking neurons should have certain properties[10]. In particular, a neuron that is part of a population should possess a tuning curve: on average, it should spike more frequently for a narrow range of input values, to which it is tuned[16,17]. Figure 1c shows data from[18] corresponding to spike rate measurements of a single neuron in vivo. The corresponding tuning curve has a bell-shape dependence on the drift direction of the input visual stimulus. The measured neuron spikes more frequently when the drift direction is around −20°: it is in charge of representing the input over a narrow range of angles. In general, all neurons in a given population have similar tuning curves of rate versus amplitude. However, the tuning curves are shifted and distributed in order to cover the whole range of input amplitudes. The ensemble of tuning curves in the population then forms a basis set of functions (bottom panel of Fig. 1e), similar to the sines and cosines of a Fourier expansion[10,19].

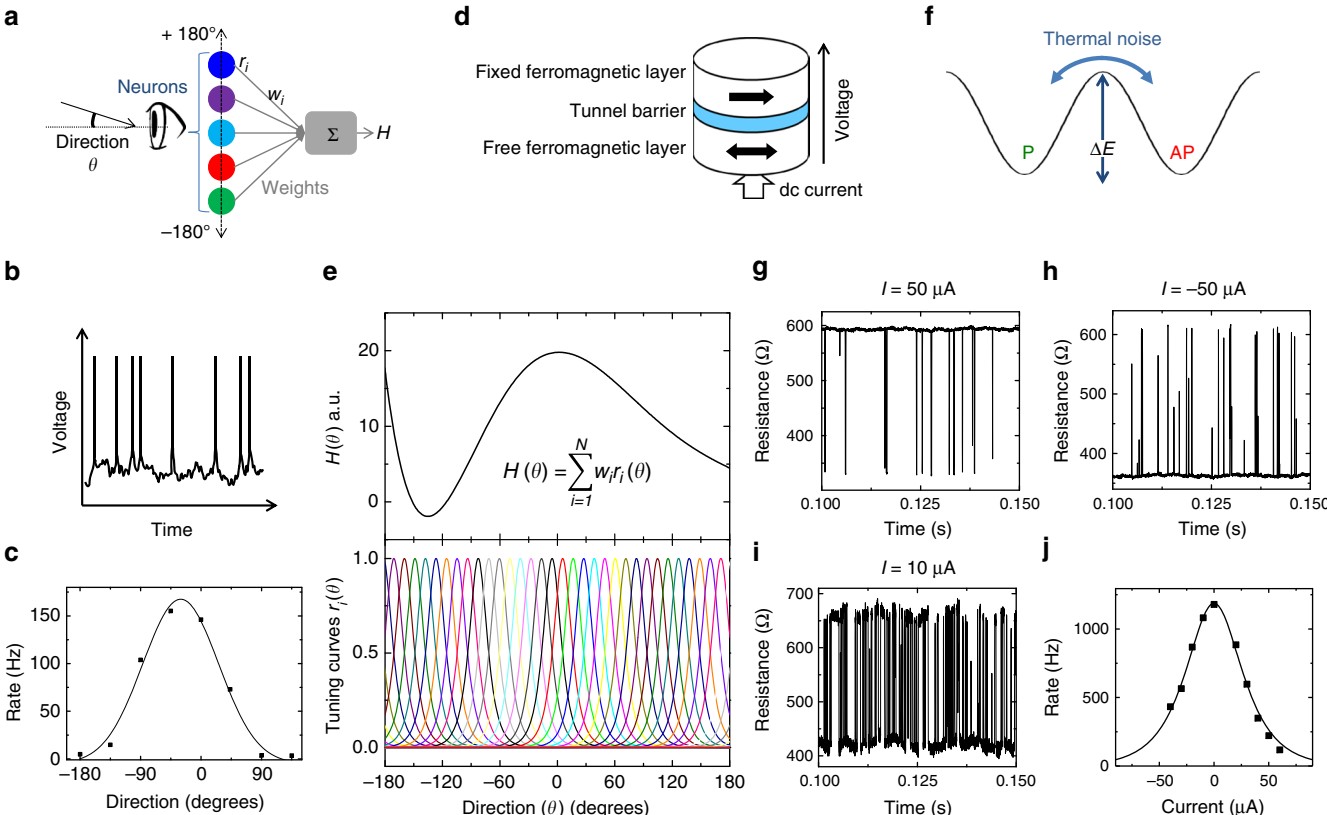

**Fig. 1** Neural and superparamagnetic tuning curves for population coding. **a** Schematic representing information reconstruction from a population of neurons. Each neuron (color dots) senses a specific range of stimuli (orientations). The represented function $H$ is computed as a weighted sum of the rates of each neuron. **b** Sketch of a typical neuron firing pattern. The emitted voltage is plotted versus time. **c** Tuning curve of a neuron: spiking rate versus direction of the observed target, reproduced with data from[18]. Experiment (symbols) and Gaussian fit (solid lines) are shown. **d** Schematic of a superparamagnetic tunnel junction. **e** Polynomial function constructed from a weighted sum of the tuning curves of the population of neurons. **f** Energy landscape of the magnetic device. **g–i** Experimental measurements of the resistance versus time of a superparamagnetic tunnel junction for injected currents of 50 μA (**g**), −50 μA (**h**), and −10 μA (**i**). **j** Rate of the superparamagnetic tunnel junction versus current. The experimental results (symbols) and analytical fit (solid line) are shown

In the present work, we show that a nanodevice—the superparamagnetic tunnel junction—naturally implements neurons for population coding, and that it can be exploited for designing systems that can compute and learn. The behavior of the nanodevice directly provides a tuning curve and resembles a spiking neuron. Without the use of explicit analog-to-digital converters it transforms an analog input into a naturally digital output that can then be processed by energy-efficient digital circuits, resulting in a low area and low energy system. The spiking nature of the neurons gives a stochastic character to the system, which appears a key element of its energy efficiency and a source of robustness.

After having studied and modeled the tuning curve provided by superparamagnetic tunnel junctions, we demonstrate experimentally that they can be assembled to implement a physical basis set of expansion functions and carry out computations. We simulate larger systems composed of several populations of superparamagnetic junctions and show that they can be combined in order to learn complex non-linear transformations, and that the resulting systems are particularly resilient. We propose and evaluate an implementation associating the nanodevices with conventional CMOS (complementary metal oxide semiconductor) circuits, highlighting the low area and energy consumption potential of the approach.

## Results

**Tuning curve of a superparamagnetic tunnel junction**. Magnetic tunnel junctions, schematized in Fig. 1d, are devices composed of two ferromagnets: one with a fixed magnetization and the other with a free magnetization that can be either parallel (P) or antiparallel (AP) to the fixed magnet. Large junctions are stable and used today as non-volatile memory cells in spin-torque magneto-resistive random access memories (ST-MRAM)[20]. However, when the junctions are scaled down, the energy barrier confining the magnetization in the P or AP states ($\Delta E$ in Fig. 1f) is reduced. For very small lateral dimensions of the junctions (typically below a few tens of nanometers), thermal fluctuations can destabilize the magnetic configuration, generating sustained stochastic oscillations between the P and AP states[21–23] (Fig. 1f). This phenomenon, called superparamagnetism, leads to telegraphic signals of the resistance as a function of time through magneto-resistive effects. These stochastic junctions have recently attracted interest for novel forms of computing[22,24,25]. Here, we experimentally study superparamagnetic junctions with a $Co_{27}Fe_{53}B_{20}$ magnetic switching layer of thickness 1.7 nm, and an area of $60 \times 120$ nm$^2$ (see Methods for details). Figure 1g–i shows experimental time traces of a superparamagnetic junction resistance as a function of time. The thermally induced random resistive switches follow a Poisson process[21,23,26]. This phenomenon presents similarities with the highly stochastic neural firing illustrated in Fig. 1b, also often modeled as a Poisson random process[22,23].

We propose to combine the thermally induced resistive switches arising in nanoscale magnetic tunnel junctions with spin-torque phenomena to emulate the tuning curves of stochastic spiking neurons. Indeed, when a direct current is applied across a superparamagnetic tunnel junction, the escape rates of the Poisson process are modified through spin-transfer torque (STT)[21,27]. As observed in Fig. 1g, a positive current stabilizes the anti-parallel state while a negative current stabilizes the parallel state (Fig. 1h), resulting in reduced switching rates in both cases compared to the case where $I$ is close to zero (Fig. 1i). As a consequence, the rate of the stochastic oscillator varies with the value of the applied dc current. From such measurements, we extracted the rate $r$ of the junction at various current values. The resulting experimental rate versus current curve $r(I)$ is shown in

Fig. 1j. With its bell-shape, it accurately mimics the neural tuning curve schematized in Fig. 1c. Spin transfer torque theory[23] allows deriving the analytical expression of the rate of a superparamagnetic tunnel junction as a function of current:

$$r(I) = \frac{r_0}{\cosh\left(\frac{\Delta E\, I}{k_B T\, I_c}\right)} \qquad (1)$$

In Eq. 1 (derived in Methods), $k_B T$ is the thermal energy, $I$ the applied current, and $I_c$ the critical current of the junction. As shown by the solid line in Fig. 1j, this equation fits well the experimental result, with $\frac{\Delta E}{k_B T} \approx 13$, and a critical current $I_c$ of 300 µA. The natural rate $r_0 = \varphi_0 \exp\left(-\frac{\Delta E}{k_B T}\right)$ (with an attempt frequency $\varphi_0$ of 1 GHz) is the peak frequency at zero current, of the order of a few thousand Hertz in the case of the junction of Fig. 1j. Superparamagnetic tunnel junctions therefore have a well-defined tuning curve $r(I)$, which allows them to sense a narrow range of currents around zero current (here around ±50 µA). The shape of the superparamagnetic tuning curve approximates a Gaussian function, which is favorable for population coding, as the ensemble of Gaussian functions with all possible peak positions forms a well-known basis set[10].

**Population coding with superparamagnetic tunnel junctions**. Following the basic principles of ref.[10], for our approach, we need to produce a population of superparamagnetic tunnel junctions that can construct non-linear functions $H$ of its inputs through a simple weighted sum of the nanodevice non-linear tuning curves $r_i$:

$$H(\theta) = \sum_{i=1}^{N} w_i r_i(\theta). \qquad (2)$$

Non-linear transformations underlie a wide range of computations such as pattern recognition, decision making or motion generation[19,28–32]. For example, navigating in a crowded room requires generating complex trajectories to avoid obstacles. The top panel of Fig. 1e displays an instance of such a trajectory produced through Eq. 2 using the basis set formed by the tuning curves in the bottom panel. These outputs are generated by the ensemble of the neural responses. Therefore, having a full population rather than a single superparamagnetic tunnel junction allows for parallel processing of each neuron, as well as resilience to failure of the devices (see Supplementary Notes 2 and 3 as well as Supplementary Fig. 2 and 3). In addition, the population outputs correspond to time averages of the stochastic neural firing patterns, which make them robust to noise. Good approximations of these output curves can be obtained quickly and at low energy by averaging the first few observed spikes, whereas more precision can be gained by increasing the measurement length.

To build a population, we need to tune each junction to different ranges of input currents. An elegant solution for this purpose is to leverage a spintronic effect called spin-orbit torques (as detailed in Supplementary Note 1 and Supplementary Fig. 1)[33–35]. However, shifting the tuning curves can also be achieved by applying individual current biases $I_{bias}$ to each junction, so that the effective current $I_{eff}$ flowing in a junction is shifted compared to the common applied current $I_{app}$: $I_{eff} = I_{app} - I_{bias}$. This method has been used in CMOS-only hardware implementations of population coding[12]. Figure 2a shows the normalized rates $r/r_0$ of an experimental population of nine junctions obtained with this method (symbols) and the corresponding fits with Eq. 1 (solid lines). We have chosen the shifts so that the junctions in the population cooperate to sense a large range of currents

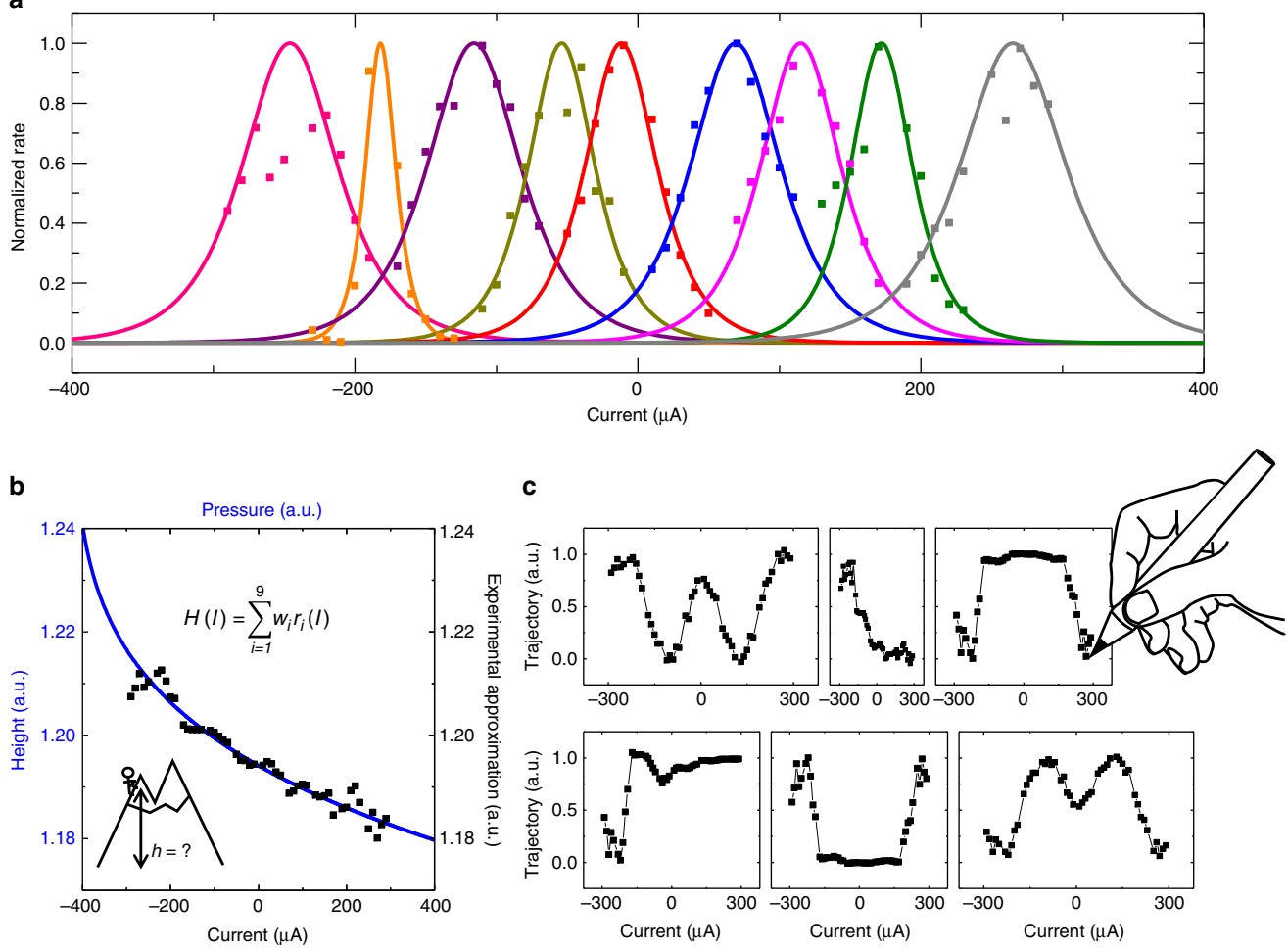

**Fig. 2** Representing non-linear functions with superparamagnetic tunnel junctions. **a** Rates versus current for nine superparamagnetic tunnel junctions with shifted tuning curves. Symbols correspond to experimental data while solid lines are analytical fits with Eq. 1. The switching rate of each junction is normalized by its natural rate $r_0$. **b** Example of the altimeter sensor. The solid blue line corresponds to the barometric formula, converting an air pressure measurement into the local height. The black symbols correspond to the experimental approximation of this function generated with Eq. 3, using the basis set data from **a** and performing the weighted sum with a computer. **c** Six examples of cursive letters (w, i, n, r, u, m) generated from the experimental junction tuning curves of **a** following the same procedure as in **b**

between −300 and +300 μA. As can be observed in Fig. 2a, the junctions are not identical due to the polycrystalline nature of the free ferromagnetic layer (see Methods). This variability affects both the critical current $I_c$ and the energy barrier $\Delta E$, resulting in the width variations of the tuning curves in Fig. 1a, but also in the variation of natural rates that for this set of junctions span from a few Hertz to 70 kHz.

Despite this variability, the experimental basis set of nine superparamagnetic tuning curves can be used to perform useful computations. We encode the input to process in the current applied to the junctions. We use the junctions measured output rates $r_i(I)$. Then this data is used to achieve the transformation to the output function $H$ by performing a weighted sum through:

$$H(I) = \sum_{i=1}^{9} w_i r_i(I) \qquad (3)$$

where the optimal weights in Eq. 3 for the desired function $H$ are obtained through matrix inversion on a computer (see Methods).

Non-linear transformations of inputs as in Eq. 3 are essential in many applications. A first field of applications is sensors, which generally require converting a measured quantity into the sought-

after information through a complex equation. For instance, a thermometer will convert the height of a column of liquid into a temperature. Similarly, an altimeter measures the local air pressure that is then converted into the corresponding height through the barometric equation shown in solid line in Fig. 2b. We have used our experimental basis set to implement this equation. As can be seen in Fig. 2b, the output reconstructed from the experimental data using Eq. 3 (symbols) reproduces the desired function. Another application making substantial use of non-linear transformations is motor control. Indeed, directing robotic arms, guiding vehicles or moving biological fingers require the generation of complex trajectories. For instance, we use here our superparamagnetic basis set to create handwriting. Figure 2c shows that we can successfully output six letters, which means that our small experimental system of nine junctions could potentially guide a robot's arm to write. These results constitute the proof of concept of computing with electronic nanodevices through population coding.

**A computing unit that can learn**. We have seen that the benefit of representing a value, such as the current $I$, by a basis set

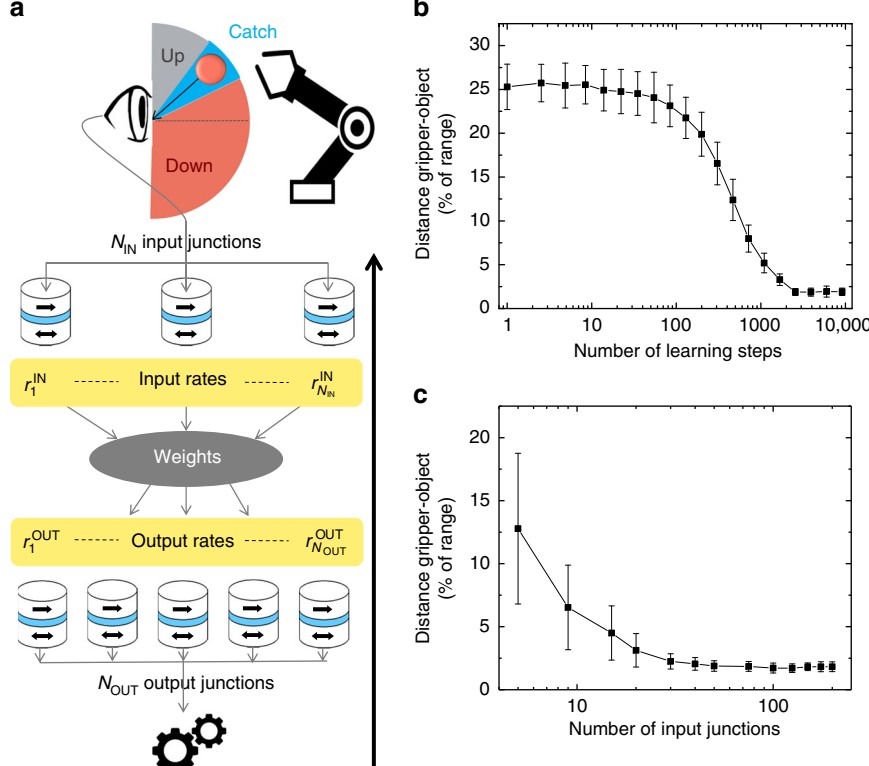

**Fig. 3** Learning to transfer information between two interconnected populations. **a** Schematic of the system and associated learning process. **b** Distance between the gripper and the object (i.e., grasping error) versus the number of learning steps (populations of 100 junctions). **c** Distance between the gripper and the object after 3000 learning steps as a function of the number of junctions in the input population. The output population has 100 junctions. For all figures, each data point corresponds to the average over 50 trials and the error bar to the associated standard deviation

population is that non-linear transformations on this value, $H(I)$, can be conducted by operating only linear operations. However, in order to realize multi-step computations, series of non-linear transformations are necessary. As a consequence, the result $H(I)$ of the first transformation should be represented by a basis set as well, implemented by an output population.

For this purpose, we can take inspiration from biology, where neurons in different populations are densely connected through synapses which control the strength of the connection. This configuration has indeed multiple advantages. In particular, the weight values can be learnt from example data, and the high degree of interconnection provides a high resilience to noise and variability in the synapses and neurons. In neuroscience models, the rates of an output population are linked to the input rates through linear weights $w_{ij}$[11,25]:

$$r_j^{OUT} = \sum_{i=1}^{N} w_{ij} r_i^{IN} \quad (4)$$

The encoded value $Y$ can then be determined by counting the switching rates of the output population: $Y$ is equal to the mean of the values of the stimulus to which the neurons are tuned, weighted by the spiking rates of the corresponding neurons[10,29]:

$$Y = \frac{\sum_{j=1}^{N} I_{\text{bias}j} r_j^{OUT}}{\sum_{j=1}^{N} r_j^{OUT}} \quad (5)$$

The error of the system is then the distance between $H(I)$ and $Y$.

To evaluate this approach before designing the full system, we perform numerical simulations of transformation learning with two populations of superparamagnetic tunnel junctions (see Methods). We choose parameters for the junctions that reflect the experimental values and variability of their energy barrier and their critical current.

We first focus on an example of a sensory-motor task (illustrated in Fig. 3a) to explain our system and demonstrate the transfer of information between two basis sets, implemented by two different populations. A robot observes an object with a visual sensor and attempts to grasp it with a gripper. The input population of junctions receives a current $I$ encoding for the orientation of the object. The output population represents the orientation $Y$ of the gripper. We want to find the weights $w_{ij}$ allowing for the orientation $Y$ of the gripper to match the orientation of the object, and show how they can be learned. For this purpose, we follow an error and trial procedure, similar to the one described in ref.[36]. Originally, the weights are random. At each trial, the object is presented at a different orientation and the weights are modified depending on the success of the grasping (see Methods for quantitative details on weights modifications):

If the gripper succeeds—i.e., if its orientation is close enough to the orientation of the object to be in the catch zone—the weights are unchanged.

If the gripper strikes in the up zone, the synaptic weights connecting the sensor network to motor junctions which are tuned to orientations above (resp. below) of the gripper are decreased (resp. increased).

If the gripper strikes in the down zone, the opposite is implemented.

The key advantage of this learning rule is its simplicity: there is no need to perform a precise measurement of the error (here distance between the gripper and the object) as required by most learning methods in the literature[37,38]. Note that the proposed system is independent of this learning rule and that different algorithms could be used to perform more complex tasks. Figure 3b shows that the distance between the object and the gripper is progressively decreased through repeated learning steps. After 3000 learning steps, the mean error is below 2.5% of the range: learning is successful. As can be seen in Fig. 3c the grasping error decreases as the number of junctions in the input population increases. The precision of the result indeed improves as the population grows, better approximating an ideal, infinite

basis set. Figure 3c also demonstrates that transfer of information between populations of different sizes can be achieved, allowing changes of basis if needed.

The example of the gripper in Fig. 3 shows how we can transfer information without degradation from one population to a different one performing a basis change. Now we show that our system and our simple learning procedure can also transform information during the transfer between populations, in other words, realize more complex functions than the identity of Fig. 3. In Fig. 4a, we illustrate increasingly more complicated transformations: linear but not identity (double), square, inverse, and sine of the stimulus. Each can be learned with excellent precision, similar to the identity.

Furthermore, by adding another matrix of synaptic weights and another population of junctions after the output of our system, we can realize transformations in series (see Methods, Supplementary Note 4 and Supplementary Fig. 4). An example of this is shown in Fig. 4a, as indicated by the label Series, where the square of the sine is performed.

The system can also be adapted for learning and performing tasks involving several inputs. A possible solution to process multiple inputs with a population is to combine them in a single input that can then be presented to the superparamagnetic tunnel junctions, consistently with the approach recently presented in ref.[39]. Here we propose a different approach where each input is sent to a different input population, and the rates originating from these separate populations are combined into a single neural network (see Methods, Supplementary Note 4 and Supplementary Fig. 5). In this way, by using several populations as inputs and outputs, multi-input multi-output computations, and therefore transformations in several dimensions can be learned. In particular, we used this approach to learn the conversion of coordinates from polar to Cartesian system. The results corresponding to this task are labeled 2 inputs in Fig. 4a.

The excellent precision of these transformations, obtained with junction parameters and variability extracted from experiments, demonstrate the resilience of our system to variability. Additional simulations reported in Supplementary Note 2 and Supplementary Fig. 2 indicate that variability of the critical current barely affects the system. Figure 4b shows the distance between the object and the gripper as a function of the variability on the energy barrier (and thus on the natural rate). The level of variability corresponding to experiments is indicated. We observe that even larger levels of variability can be tolerated by the system, which is promising for realizing population coding with ultra-small junctions despite lithographic defects.

Finally, it should be noted that scaling down the junctions allows decreasing the energy consumption of a population to tens of picoJoules, as show on Fig. 4c (see Methods). Furthermore, as typical in stochastic computing systems[40], the precision of the system is

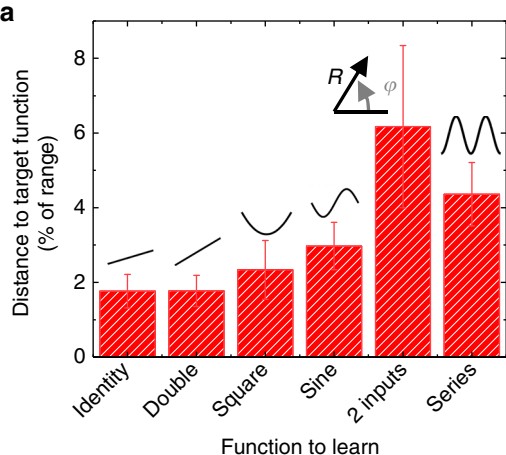

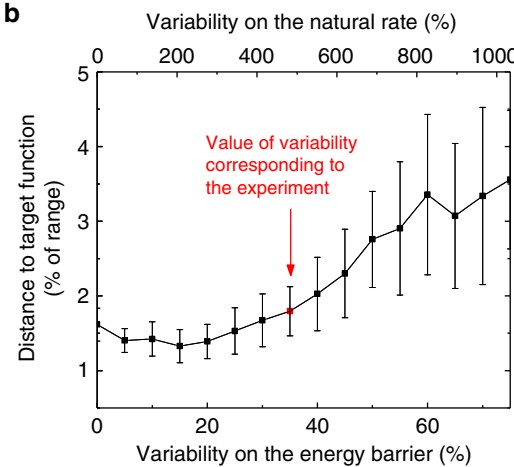

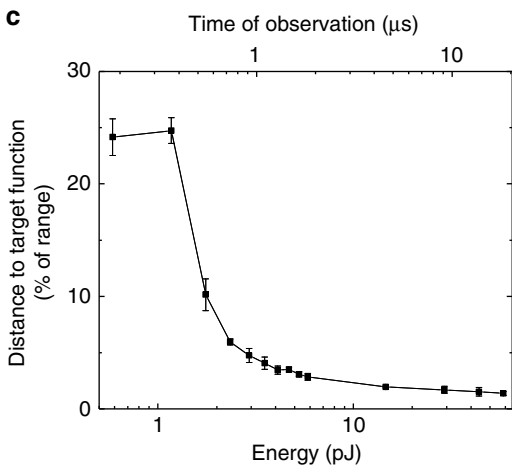

**Fig. 4** Evaluation of stochastic population coding with superparamagnetic tunnel junctions. **a** Performance of several transformations, including non-linear. The 2 inputs label corresponds to transformation from polar to Cartesian coordinates. The series label corresponds to two transformations in series implementing the function $\sin^2(x)$. **b** Distance to target versus variability of the energy barrier (bottom axis) and variability of the natural frequency (top axis). The experimental variability is indicated in red. **c** Distance to target for different times of observation during which switching rates are recorded, leading to different energy dissipated by the junctions (see Methods). Longer acquisition time allows better precision of the transformation, but leads to higher energy consumption. Each population is composed of 100 junctions and 3000 learning steps are used. Each data point corresponds to the average over 50 trials and the error bar to the associated standard deviation

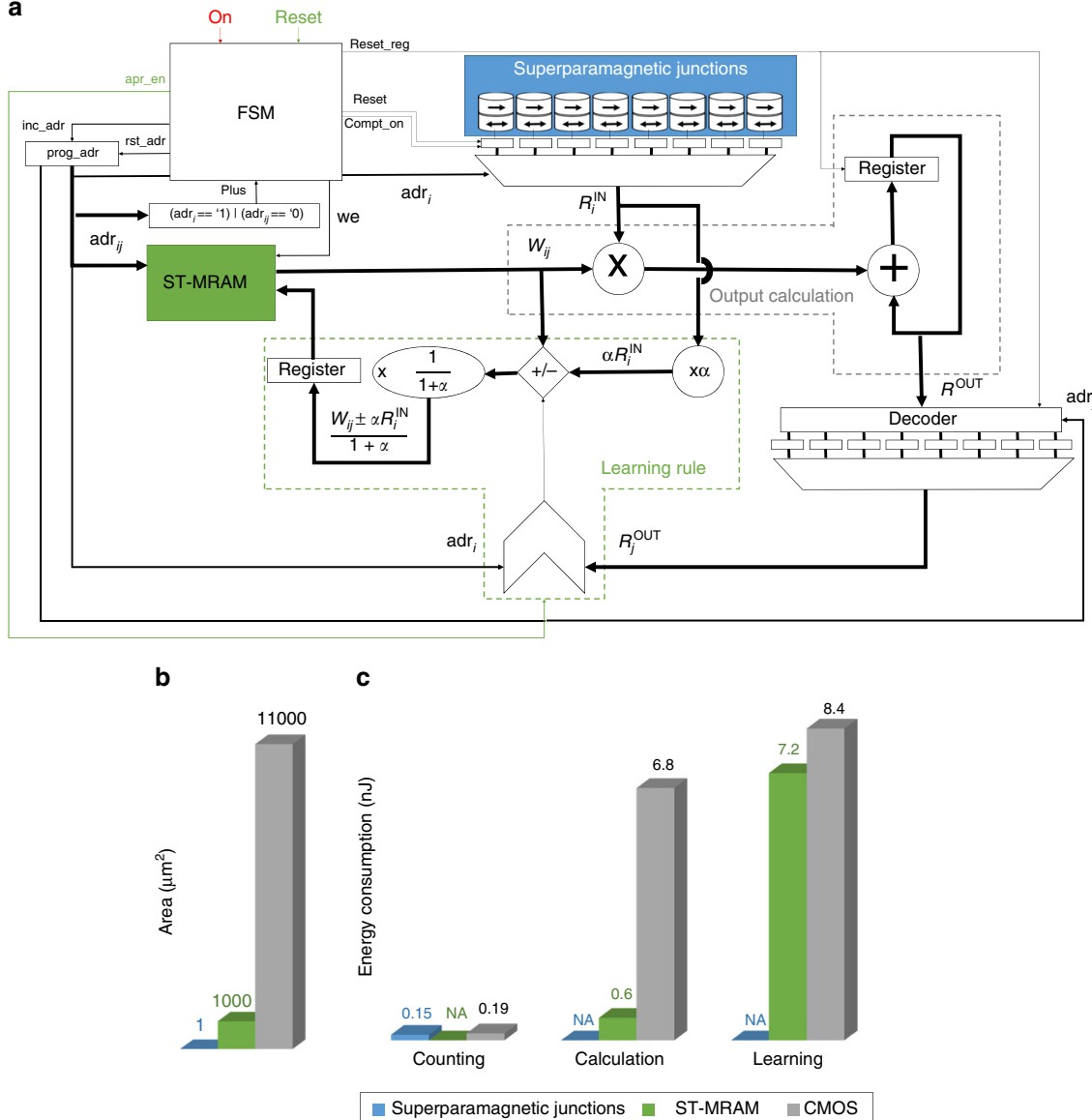

**Fig. 5** Design of the full system for the gripper task. **a** Schematic illustrating the data path of the designed system, associating CMOS circuits, superparamagnetic tunnel junctions, and stable magnetic junctions used as MRAM. (FSM finite state machine, WE write enable, ADR address). **b** Circuit area occupied by the superparamagnetic tunnel junctions, CMOS, and MRAM. **c** Energy consumption of the superparamagnetic tunnel junctions, CMOS, and MRAM required to perform one system operation: running the junctions, computing the output, and adjusting the weight. The system has 128 inputs and 128 outputs

directly dependent on the observation time and thus on the consumed energy, allowing choice in a precision-energy tradeoff.

**Design of the full system**. To evaluate the viability of the approach, we designed a full system associating superparamagnetic tunnel junctions as input neurons, CMOS circuits, and standard magnetic tunnel junction used as ST-MRAM to store the synaptic weights $w_{ij}$. These stable junctions can be fabricated using the same magnetic stacks as the superparamagnetic junctions (but a different sizing). The CMOS parts of the circuit were designed using standard integrated circuit design tools and the design kit of a commercial 28 nm CMOS technology (see Methods). A simplified representation of the system is shown in Fig. 5a.

The system features an ensemble of superparamagnetic tunnel junctions, to which the stimulus is applied using the current shift method introduced earlier. For the system, we assumed that the

superparamagnetic junctions were scaled to nanometer sizing (see Methods). Junctions switching events are detected by a CMOS circuit to determine the rates $r_i$. It consists of a synchronous low-power comparator, which compares the voltage across a junction and the corresponding voltage on a reference resistance (see Methods), as well as a digital edge detection logic. Each junction is associated with a digital counter counting the switches. After a stimulus operation phase, the system can compute its output using Eq. 4 using integer arithmetic. This is done by a digital circuit that we designed and is described in Methods. The synaptic weights $w_{ij}$ are stored in stable magnetic tunnel junctions sized to a 28 nm technology (see Methods). If the system is in a learning phase, the learning rule is then applied by a digital circuit, also described in Methods, which reprograms some ST-MRAM cells. A more detailed presentation of the data path and of the operation of the system is presented in Supplementary Note 5 and Supplementary Fig. 6 and 7.

It is also a possibility to design the system using a single superparamagnetic junction, and to implement the population response through time multiplexing. This approach would allow avoiding the effects of device variability. However, it would also increase conversion time by the number of input neurons, giving a very low bandwidth to the system. As the superparamagnetic junctions have low area and the system features a natural resilience to device variability, we propose to physically implement the population with an actual population of junctions.

As presented the circuit features a single input. As explained earlier, it may be extended to several inputs, following the principle presented in Supplementary Note 8 and Supplementary Fig. 11.

Figure 5b shows the circuit area occupied by superparamagnetic junctions, CMOS, and ST-MRAM on chip, for a system with 128 inputs and 128 outputs. The total area is very low (12,000 μm²) showing that the concept is adapted to be used in low-cost intelligent sensor applications. The area is dominated by the CMOS circuits, while the area occupied by the superparamagnetic junctions is negligible.

Figure 5c shows the energy consumption to perform the gripper task, for one operation of the gripper, separating the three phases (observation of the stimulus, computation of Eq. 4, learning), and the three technologies present in the system. Results concerning systems with different numbers of inputs and outputs are presented in Supplementary Note 6 and in Supplementary Figs. 8, 9 and 10. The total energy is very low: 23 nJ during the learning phase, and 7.4 nJ when learning is finished.

It is instructive to compare these results with solutions where neurons would have been implemented with purely CMOS circuits. A detailed comparison to four different approaches is presented in Supplementary Note 7 and Supplementary Table 1. A natural idea is to replace our junctions and their read circuitry by low-power CMOS spiking neurons, such as those of ref.[41], which provides features similar to our nanodevices (analog input and spiking digital output). This strategy works but has high area requirements (higher than 1 mm²), and would consume more than 330 nJ per operation. Alternative options rely on analog computation, for example exploiting neurons such as ref.[13]. Such solutions require the use of an explicit analog to digital conversion (ADC), which actually becomes the dominant source of area and energy consumption. Even extremely energy efficient ADCs[42] require a total of 20 nJ/conversion and an area of 0.2 mm². Finally, a more conventional solution, using a generic processor and not an application-specific integrated circuit would have naturally used order-of-magnitudes more energy.

The low-energy consumption of our system arises from a combination of three major factors. The superparamagnetic junctions consume a negligible energy (150 pJ), and allow avoiding the ADC bottleneck present in other approaches by implementing a form of stochastic ADC in a particularly efficient manner. The use of a stochastic approach and of integer arithmetic in the CMOS part of the circuit is particularly appealing in terms of energy consumption. Finally, associating both CMOS and spintronic technology on-chip limits data transfer-related energy consumption.

## Discussion

In this work, we show that superparamagnetic tunnel junctions are promising nanodevices for computing in hardware through population coding. We experimentally demonstrate that these components intrinsically mimic the tuning curve of neurons through their non-linear frequency response to input currents. We realize a basis set of expansion functions in hardware from a small population of junctions, and show how they can encode information and compute by generating complex functions such as letters. Using a physical model of the superparamagnetic tuning curves, we demonstrate that combined populations of junctions can learn non-linear transformations with accuracy, even with substantial device-to-device variability. Our system acts as a stochastic computing unit that can be cascaded to perform complex tasks. The design of the full system associating the junctions with CMOS circuits and ST-MRAM shows the potential of the approach for extremely low-area and low-energy implementation.

Our work reproduces the essence of population coding in neuroscience, with some adaptations for implementation with nanoelectronics. In population coding theory, neuronal correlation[11,43], the meaning of the time[11], as well as decoding techniques[43] are contentious topics. In our system, these aspects were guided by the properties of the nanodevices and by circuit design principles. The input neurons spike in an uncorrelated fashion, as their noise originates from basic physics. The time is divided into discrete phases, allowing the use of counters, and finite state machines in the system. The information is decoded by counting spikes using simple unsigned digital counters.

It is also important to note that in our system, the junctions act as a form of spiking neurons that employ rate coding, similarly to several population coding theories[10,11]. The spiking nature of the neurons offers considerable benefits to the full system: it naturally transforms an analog signal into easy-to-process digital signals. The stochastic nature of the neurons is one of the keys of the energy efficiency and of the robustness of the system. It also gives the possibility for the system to provide an approximate or precise answer depending on the time and energy budget, similarly to stochastic computing[40,44]. The rest of the system is rate based, which allows learning tasks in a straightforward manner. Another possibility would have been to perform the entire operation in the spiking domain, as is common in the neuromorphic engineering community[45–47]. However, learning in the spiking regime remains a difficult problem today[48], and involves more advanced concepts and overheads[47]. Therefore, our system is designed to take benefits from both the spiking and the rate-coding approaches.

In summary, our system mixes biological and conventional electronics ideas to reach low-energy consumption in an approach that might presage the future of bioinspired systems. Our results therefore open the path to building low energy and robust brain-inspired processing hardware.

## Methods

**Experiments**. Samples: The samples are in-plane magnetized magnetic tunnel junctions. They were fabricated by sputtering, with the stack: substrate (SiO₂)/ buffer layer 35 nm/IrMn 7 nm/CoFe 2.5 nm/Ru 0.85 nm/CoFeB 2.4 nm/MgO-barrier 1.0 nm/CoFeB 1.7 nm/capping layer 14 nm. The whole stack was annealed before microfabrication at 300 °C under a magnetic field of 1 Tesla for 1 hour. Patterning was then performed by e-beam lithography, resulting in nanopillars with elliptic $60 \times 120$ nm² cross-sections.

Measurements: The measurements are performed under a magnetic field that cancels the stray field from the synthetic antiferromagnet. In Fig. 2a the curves, initially centered on zero voltage, have been shifted along the $x$ axis (current).

Analytical fits: Equation 2 was used for the analytical expression of the frequency of the junctions. The parameters $\Delta E$ and $I_c$ were chosen for each junction so to fit best the experimental data. The parameters used are (from left to right in Fig. 1):

$\Delta E/k_B T$ = 16.5, 8.87, 18.58, 17.92, 12.95, 18.675, 11.75, 18.35, 12.14
$I_c$ ($A$) = 5e-4, 8.5e-5, 5.5e-4, 3.8e-4, 2.96e-4, 5.35e-4, 3e-4, 3.6e-4, 4.1e-4

Variability: The variability in the parameters stems from the polycrystalline structure of the free ferromagnets. Instead of a full layer reversal, only a fraction of the ferromagnet switches back and forth. This explains why junctions of this size are unstable and why their parameters vary strongly from device to device.

Finding the weights by matrix inversion: Obtaining Fig. 2 requires using appropriate weights. Equation 3 can be rewritten as $H = wR$, where $w$ is the line vector of the weights and $R$ the matrix of the rates where each column corresponds

to a junction and each line to a particular current. In consequence, the weights can be found analytically by $w = HR^{-1}$. Here the weights are found using the experimental values for $H$ and $R$.

In Figs. 3 and 4, the weights are obtained by the learning process and no matrix inversion is necessary.

Barometric formula: The height is given by $z = z_1 + T_0/A \times [1 - (p/p_1)^{1/\alpha}]$ where $\alpha$ is 5.255, $A/T_0 = 2.26 \times 10^{-5}$ and $p_1$ and $z_1$ are chosen measure points. Here we use height $= 1 + 0.3 \times (1 - ((I_{dc} + 4.2 \times 10^{-4})/0.1)^{1/\alpha}))$ as target function.

**Numerical simulations**. Choice of the parameters and variability: For the energy barrier $\Delta E$ we use a uniform distribution, centered around 13.78 $k_B T$ and of span of 9.65 $k_B T$ (0.35% variability). This corresponds to a natural rate of 518 Hz.

For the critical voltage $V_c$, we use a Gaussian distribution of mean 0.142 V and standard deviation 0.037 V (0.26% variability).

These parameters correspond to those extracted from the experiment.

Simulations of a population of junctions: In our simulations, we chose to control the junctions by voltage, which makes it easy to apply one common stimulus to all junctions. The behavior of the junctions is modeled by a two-state Poisson process. The stimuli received by the junctions modify the escape rates of each state of the process.

$$\varphi_{P/AP} = \varphi_0 \exp\left(-\frac{\Delta E}{k_B T}\left(1 \pm \frac{V_{eff}}{V_c}\right)\right) \qquad (6)$$

In the case of the input population, $V_{eff} = V - V_0$ where $V$ is the stimulus common to all junction and $V_0$ is voltage to which the considered junction is tuned. $V_c$ is the critical voltage. The probabilities for the junctions to switch during a time interval d$t$ are:

$$P_{P/AP} = 1 - \exp\left(-dt\varphi_{P/AP}\right) \qquad (7)$$

The numerical simulations are run as follows: At every time step d$t = 439$ μs and for each junction the probability to switch state is computed and a random number is generated to decide if the switch occurs. After 100 time steps, the frequency of each junction is computed[49].

Interconnecting the two populations of junctions: We seek to connect the two populations of junctions so that the rates of the output junctions obey Eq. 4. To do so we inverse Eq. 2 to compute the voltage to be applied to each output junction so that its rate satisfies Eq. 4. We then simulate the population of output junctions as described above. Here $V_{eff}$ correspond to the computed voltage.

Stimulus range covered by the junctions: The input population of junctions is assembled so that it can sense voltages over a range spanning here from $-0.15$ to $+0.15$ V. This range thus encodes the possible orientations of the observed object. Shifting the rates of the junctions in different ways allows for sensing different ranges, as will be seen for the coordinate transformations.

Learning rule: For all the output junctions $j$ for which the connections to the input population should be increased, the weights are modified as follows:

$$\forall i \in [1, N_{IN}], W_{ij} \leftarrow \left(W_{ij} + \alpha \frac{r_i^{IN}}{r_0}\right)\frac{1}{1+\alpha} \qquad (8)$$

For all the output junctions $j$ for which the connections to the input population should be decreased, the weights are modified as follows:

$$\forall i \in [1, N_{IN}], W_{ij} \leftarrow \left(W_{ij} - \alpha \frac{r_i^{IN}}{r_0}\right)\frac{1}{1+\alpha} \qquad (9)$$

$r_0$ is the natural rate of the junctions and $\alpha$ is the learning rate. Low values of $\alpha$ slow down the learning, while high values of $\alpha$ fasten the learning but limit its performance. Here we found the value 0.001 to be appropriate for $\alpha$.

Measure of the error: The error is the absolute value of the difference between the orientation of the target and the orientation given by the output junctions to the gripper. It is expressed as a percentage of the range of possible orientations (here from $-0.15$ to $+0.15$ V). It is computed as an average over 50 randomly chosen trials.

One-dimension coordinate transformations: The task is performed in the same way as in the catching target case, with the orientation of the object $Z$ being replaced by the result of the transformation operation $T(Z)$. The distance gripper-target is computed as the absolute difference between the expected value of the transformation $T(Z)$ and the numerically computed value. It is expressed as a percentage of the range of possible expected values. For identity ($T(Z) = Z$) and double ($T(Z) = 2Z$), the stimulus range is $-0.15$ to $+0.15$ V. For square ($T(Z) = Z^2/0.15$) the stimulus range is $-0.15$ to $+0.15$ V. For sine ($T(Z) = \sin(Z \pi / 0.15)/$ 0.15), the stimulus range is $-0.15$ to $+0.15$ V.

Two-dimensional coordinate transformation: Here the transformations to perform are $x = R\cos(\varphi \pi/0.6)$ and $y = R\sin(\varphi \pi/0.6)$.

The stimulus ranges are 0 to 0.3 V for R and 0 to 0.3 V for $\varphi$. The range for both $x$ and $y$ is 0 to 0.3 V. Four populations of junctions encode the four coordinates R, $\varphi$, $x$, and $y$.

The two input populations R and $\varphi$ are concatenated into a single population. Its number of junctions is the sum of the number of junctions in each population $N_{IN} = N_R + N_\varphi$. Two weights matrices ($W_x$ and $W_y$) connect the input (R, $\varphi$) to the ouput junctions ($x$, $y$). The weights matrices $W_x$ and $W_y$ have the dimensions $N_x \times N_{IN}$ and $N_y \times N_{IN}$. Where $N_x$ ($N_y$) is the number of junction encoding $x$ ($y$). Learning of the weights is implemented as described previously.

The distance gripper-target is computed as the absolute 2D distance between the target and the gripper and is expressed as a percentage of the range for $x$ and $y$.

Transformations in series: Here we want to perform the square of the sine ($T(Z) = (\sin(Z \pi/0.15)/0.15)^2$) in two successive steps. We have three populations of superparamagnetic junctions. The middle population is connected to the input population by a weight matrix $W_1$ and the output population is connected to the middle population by a weight matrix $W_2$. $W_1$ and $W_2$ are trained as in the single transformation case, so that they respectively perform the sine and the square transformation.

**Energy consumption of a population**. Power/energy dissipated by the superparamagnetic junctions: We consider scaled down junctions with parameters $\Delta E = 6k_B T$ and $V_c = 0.1$ V, shifted by individual voltage biases between $-0.1$ and 0.1 V. This corresponds to a natural rate of 1.23 MHz.

The power consumption due to the shifting is

$$P_{shift} = \sum_{i=1}^{N} \frac{V_{shift}^2}{R} \qquad (10)$$

where $N = 100$ is the number of junctions, $V_{shift}$ is the maximal firing voltage for the i-th junction, and $R$ is the resistance of the junctions.

For a $RA = 20$ μΩ $\times$ cm$^2$ and a $d = 7.7$ nm diameter the resistance is $R = 424$ kOhm.

The power consumption is $P_{shift} = 0.8$ μW.

The maximal power consumption for the stimulus is $P_{stim} = N \times 0.1^2/R = 2.4$ μW.

So the total power is $P = 3.2$ μW.

The distance to the target function shown in Fig. 4c is computed through the same numerical simulation as in the experimental parameter case. Here the time step is d$t = 183$ ns.

The energy consumption is the power $P$ multiplied by the duration of the observation.

**Design of the full system**. The full system was designed and its performances were estimated using standard integrated circuit design tools developed by the Cadence corporation (Virtuoso, Specter, RTL Compiler, ncsim and Encounter), associated with the design kit of a commercial low power 28 nm technology.

The CMOS digital parts of the system were designed with the Verilog description language at the register transfer level, and synthesized to the standard cells provided with the design kit with Cadence RTL Compiler. Overall, the circuits were optimized for low-area and low-energy consumption, and not for high speed computation. Their area was estimated using the Cadence Encounter tool. For estimating their energy consumption, value change dumps files corresponding to the gripper task were generated using Cadence ncsim and the power consumption was estimated using Cadence Encounter.

The superparamagnetic junctions were modeled based on the previous Methods section, assuming[50] $d = 11$ nm diameter, a size that has been demonstrated experimentally. The energy consumption for the detection of the spikes was based on Cadence Spectre simulation of a simple circuit, presented in Supplementary Note 5, Supplementary Fig. 7, and based on the stimulus value corresponding to the highest energy consumption. The stimulus is applied to reference resistors whose resistance is intermediate between the parallel and anti-parallel state resistance of the superparamagnetic tunnel junctions, as well on the superparamagnetic tunnel junction. At each clock cycle, the voltage at the junction and at the reference resistor is compared by a low-power CMOS comparator (Supplementary Fig. 7). Simple logic comparing the result of the comparison to the same result at the previous clock cycle allows detecting the junction switching events, which are counted by an eight-bit digital counter. (Each junction is associated with one counter).

At the end of the counting phase, the system then computes Eq. 4 in a sequential manner, controlled by a finite state machine (Fig. 5a) described in Supplementary Note 5. The synaptic weights are stored in eight-bit fixed point representation in an ST-MRAM array. Computation is realized in fixed point using integer addition and multiplication circuits. The ST-MRAM array was modeled using assumptions in terms of area and energy consumption as expected for a 28 nm technology[51]. ST-MRAM read and write circuits are modeled in a behavioral fashion, using results of ref.[52] for evaluating their area and energy consumption.

The learning circuit can be activated after the computation phase optionally. Based on the learning rule described above, computed in fixed point representation, the ST-MRAM array is reprogrammed. In order to save energy, ST-MRAM cells

are read before programming, so that only bit that actually changed are reprogrammed (a standard technique for resistive memory[53]).

**Data availability**. The datasets generated and analyzed during this study are available from the corresponding author on reasonable request.

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

## Acknowledgements

The authors acknowledge support from the European Research Council grant NANOINFER (reference: 715872). A.M. acknowledges financial support from the Ile-de-France regional government through the DIM nano-K program. The authors thank Mark Stiles, Pierre Bessière, Jacques Droulez, and Nicolas Locatelli for fruitful discussions.

## Author contributions

D.Q and J.G. devised the study. A.F., H.K., and S.Y. designed and optimized samples. A.M. performed the measurements, theoretical analysis, and numerical simulations. T.H. designed the full system and estimated its performance. All authors analyzed the results and co-wrote the article.

## Additional information

**Competing interests:** The authors declare no competing interests.

