## [Peer Review File · Nature Communications]

Reviewers' Comments:

Reviewer #1:

Remarks to the Author:

The paper proposes to use magnetic tunnel junctions as a substrate for the realization of populations of artificial neurons, and discusses the use of these junctions for population-based encoding of information and learning. First, the paper presents an experimentally obtained "tuning curve" and explains it using spin-transfer torque theory. Subsequently, simulations are presented to demonstrate the usability of the junctions as a basis set to represent nonlinear functions, and in a learning-based scenario interconnecting two populations.

The paper proposes an interesting contribution in a promising and lively research area that could potentially enable further research on new ways how information is stored and processed. However, in my opinion, further work is needed to improve the clarity, coherence and completeness that would enable the reader to judge the scientific merit of the work and enable him or her to profit from the findings of the authors.

First and foremost, I would recommend to properly position the work into the context of artificial neuronal networks. The authors should clarify (including relevant references to literature) the relation of their work to spiking neuronal nets, spike-based vs. rate-based coding, artificial neuronal nets and other approaches. Figure 1 presents an image of a "neuron" (albeit lacking any descriptions). Do the authors aim at replacing/emulating the neuronal soma, the entire neuron with synapses, or potentially the synapses only? This should be made clear upfront. Also, the presented approach is only one of many types of population coding. Since population coding is central to the publication, a clear positioning of the work with respect to other types of population coding should be provided rather than vague/ambiguous statements ("in particular, a neuron inside a population should possess a tuning curve").

It seems rather unclear how the actual hardware implementation would look. In hardware-based emulations of neurons and synapses, the electronic components necessary to "run" the neuron are of importance, since they determine the compactness and scalability of the proposed solution. How would the firing of the neuron be detected (drop in resistance), how would be the rates computed, represented and communicated, how would the neuron be interfaced to synapses or other neurons? What is the minimum and maximum operational speed (firing rate, spike rate, ...?) of the neuron? In this respect, computer-based simulations have limited validity since some operations (e.g. matrix inversion) are computationally heavy. Again, it is very useful to outline how the proposed neurons would fit into a computational system.

The authors mention multiple times (e.g. in the first sentence of the abstract and in the conclusion) the notion of fault-tolerant information processing and device variability. Therefore, I think it would be a good idea to include representative measurements of the device variability and endurance upfront when the device is discussed (independent of the simulation-based studies), and discuss how the fault-tolerance of the population is impacted in terms independent of the simulation-based gripper study. Later on, the study can show that a break-down of some neurons or synapses (not shown) or their significantly biased performance (shown) does not hurt the overall performance of the population on a particular task.

In the gripper study, it was not clear to me whether the authors aim at presenting an improved learning algorithm on top of the more hardware-level findings ("The key advantage is ... no need to perform a precise measurement...as required by most learning methods in the literature."). Are the results of the simulation study specific for the junction-based neuronal network? It may seem the fact that increasing the size of a basis set improves a signal approximation is rather a generic finding and is intuitive. If I understood correctly, the gripper study is solely based on simulations and only using the bell-shaped model of neuronal response that is not the key contribution of the paper (Figure 3), partly with a feed-in of the STT-related parameters (Figure 4). It would have a

much larger expressive value if the neurons would be really implemented in hardware.

I would also recommend to improve the clarity of the figures and of the text. For instance, Figure 1 contains a lot of panels, some of which are actually only the background of the presented work (panels a,b,c,d,e). The first figure should summarize the key contribution in a crisp and concise way. Also, figures without descriptions (panel (a)) are not very informative. The first two paragraphs seem to me too vague at times ("the processor would melt", "the key seems to be redundancy", ...). I think the authors could make a better use of the space by providing the reader with a concise overview of background results and literature, with proper citations and terminology. Also, it might be a good idea to move some parts of the simulations (the gripper study) to Supplementary Information, since dis-proportionally large parts of the text are now occupied by its description.

Reviewer #2:

Remarks to the Author:

Prof. Grollier's group is well known for their past work on superparamagnetic MTJ's showing results like the ones shown in Fig.1f to 1j. What I believe is new experimentally is the demonstration of biased junctions with shifted tuning curves as shown in Fig.2a along with the proposal for implementing it compactly that is described in the Appendix.

This experimental demonstration plus proposal though interesting is probably not enough by itself to justify publication in Nature Communication. Presumably it is the connection to neural computing that provides this justification. Unfortunately I do not feel convinced by this part of the argument which is purely software based, with no discussion of how it could possibly be done in hardware. Let me elaborate.

The authors propose a hardware implementation of superparamagnetic tunnel junctions for basis functions for a class of problems called 'population coding'. I enjoyed reading the paper, and found the main idea interesting but I feel that there is a strong dissonance between the claims and the actual demonstration. The paper seems to justify its conclusions by drawing parallels to performance of computers for the post-CMOS era, but no actual comparisons are attempted to see how a standard CMOS implementation of their examples would have fared against this highly computer-assisted demonstration.

The abstract of the paper claims that strong requirements have until now prevented a demonstration of population coding with nanodevices. It is not clear why a standard nanoscale CMOS implementation of the present examples would have problems generating the non-linearity and stochasticity, especially considering the paper itself makes extensive use of external CMOS hardware that requires non-trivial functions as I point out below.

Currently, the abstract reads 'We demonstrate experimentally that a population of nine junctions can implement a basis set of functions, allowing, for instance, the generation of cursive letters'. I find this statement misleading. If the authors needed 9 MTJs to obtain the main experimental results of the paper (For example, Figure 2c), the demonstration would be an important step towards a novel, post-CMOS computation unit. As it stands, there seems to be an enormous amount of detail that is currently not discussed in the manuscript.

Other specific concerns and questions:

1. The starting unit of (Figure 1j) where the Gaussian rate is obtained from superparamagnetic junctions was processed by a computer. Am I correct in noting that this is currently not an experimental quantity that was measured in actual hardware? If so why is it important to achieve

the shifting (Fig.2a) in actual hardware?

2. Given the apparent difficulty in controlling the variations and randomness of the rate-response and heavy-use of external, standard hardware, one could argue that all of the results in the paper could have been obtained from a single MTJ that is shifted externally and then processed (to get the rate), multiplied and summed accordingly.

3. Figure 2a shows the processed response of 9 MTJs as a function of a current. Figure 2c shows six examples of cursive letters that are obtained by this data. Since the hardware currently does not calculate the weights, the multiplication of the weights with the basis functions, nor the summation, there is a substantial use of standard CMOS technology.

4. If the authors are envisioning a comprehensive hardware implementation, they should at least provide a sketch for implementing the weights, multiplication and summation of the basis functions.

In short, it seems to me that the basic information processing concepts discussed here are well-known and could be implemented all in software. The authors seem to be replacing an insignificant fraction of this with biased MTJ's. The rest still needs to be implemented in software.

Perhaps I am missing something. If so, perhaps the authors can clearly articulate what their contribution is ?

Reviewer #3:

Remarks to the Author:

This paper shows nanoscale magnetic tunnel junctions, which exhibits neuron-like Gaussian tuning curves. Authors have used these tuning curves for pattern recognition tasks in software. However, the reviewer has some concerns/comments as below.

1. There are some works on building CMOS circuit based on population coding:

A neuromorphic hardware framework based on population coding. IEEE International Joint Conference on Neural Networks (IJCNN).

A Low Power Trainable Neuromorphic Integrated Circuit that is Tolerant to Device Mismatch. IEEE Transactions on Circuits and Systems I: Regular Papers, 63(2), 211-221.

2. Ref [8] was not formatted correctly.

3. Sentence "Measurements of neuronal activity indicate...processed by population...rather than single.." is not entirely correct. This is just one kind of information encoding in the brain.

4. Figure 1 part (i) $I = -10\mu\text{A}$ but in the figure, it is $=0$. Needs to change.

5. Sentence "random resistive switches follow a Poisson process.." is it modeled or is it an assumption?

6. Sentence "shifting the tuning curves can also be achieved by applying..." check the below paper, where a similar idea is proposed...consider citing it.

A neuromorphic hardware framework based on population coding. IEEE International Joint Conference on Neural Networks (IJCNN).

7. Authors have shown the architecture for single input case. How the multiple inputs will be

combined in this architecture. Please elaborate.

8. The output of the nanoscale devices are spikes..how is it converted into rates using circuits. This needs to be elaborated.

9. The system seems to be a three-layer neural network. It would be helpful for readers if a system level diagram is provided where nanoscale devices could be a neuron. Also, how the weights would be trained between the layers, should be elaborated in the fig.

10. Weight modification rule is similar to a previous work as shown below:
An Online Learning Algorithm for Neuromorphic Hardware Accelerators...arXiv:1505.02495

11. The main concern is how the multi-input system will be built from the tuning curves. The reviewer has some concerns over its practical implementation. Please justify.

12. Authors should specify the advantages of the nanoscale devices over existing devices such as CMOS implementation or others such as speed and power efficiency.

We would like to thank the anonymous reviewers for their careful reading of our manuscript, and their comments that allowed us to improve the manuscript very significantly. Notably, we have now included a study of a complete system, including the CMOS circuitry associated with the nanodevices, which we hopefully strengthens our conclusions and makes this manuscript a significant addition to the field of emerging computing devices.

Below are the responses to Referee's questions and comments. We hope that this new version will have addressed all concerns raised in their remarks.

Reviewer 1

The paper proposes to use magnetic tunnel junctions as a substrate for the realization of populations of artificial neurons, and discusses the use of these junctions for population-based encoding of information and learning. First, the paper presents an experimentally obtained "tuning curve" and explains it using spin-transfer torque theory. Subsequently, simulations are presented to demonstrate the usability of the junctions as a basis set to represent nonlinear functions, and in a learning-based scenario interconnecting two populations. The paper proposes an interesting contribution in a promising and lively research area that could potentially enable further research on new ways how information is stored and processed. However, in my opinion, further work is needed to improve the clarity, coherence and completeness that would enable the reader to judge the scientific merit of the work and enable him or her to profit from the findings of the authors.

We thank the reviewer for his/her careful reading of the manuscript. To address his/her concern, we have very significantly overhauled the manuscript. In particular, the introduction has been entirely rewritten and the discussion rewritten. Our new results involving the design of a full system also allow better understanding the implications of the work.

1) First and foremost, I would recommend to properly position the work into the context of artificial neuronal networks. The authors should clarify (including relevant references to literature) the relation of their work to spiking neuronal nets, spike-based vs. rate-based coding, artificial neuronal nets and other approaches.

Our system uses both spiking aspects and rate coding. Spike-timing based approaches have attracted attention because they enable unsupervised learning through STDP. However, rate based approaches have the advantage of higher resilience to errors and stochasticity of the components. They also allow trade-off between time/energy cost and precision. The spiking nature of the input neuron is clarified up front in the introduction: *"The behavior of the nanodevice directly provides a tuning curve and resembles a spiking neuron."*

And we have now included a full discussion of the relationship with other approaches in the discussion:

"It is also important to note that in our system, the junctions act as a form of spiking neurons that employ rate coding, similarly to several population coding theories^{10,11}. The spiking nature of the neurons offers

considerable benefits to the full system: it naturally transforms an analog signal into easy-to-process digital signals. The stochastic nature of the neurons is one of the keys of the energy efficiency and of the robustness of the system. It also gives the possibility for the system to provide an approximate or precise answer depending on the time and energy budget, similarly to stochastic computing^{43,46}.

The information extracted from the neurons is processed using a relatively conventional state-based artificial neural network, but using digital integer arithmetic, as it is extremely efficient in CMOS. Another possibility would have been to perform the entire operation in the spiking domain, as is common in the neuromorphic engineering community⁴⁷⁻⁴⁹. However, relying on a conventional artificial neuron network allows us to provide the system with an online learning feature easily, while learning in entirely spiking neural network involves more advanced concepts and overheads⁴⁹.

Our system therefore mixes biological and conventional electronics ideas to reach low energy consumption in an approach that might presage the future of bioinspired systems. Our results therefore open the path to building low energy and robust brain-inspired processing hardware. ”

2) Figure 1 presents an image of a "neuron" (albeit lacking any descriptions). Do the authors aim at replacing/emulating the neuronal soma, the entire neuron with synapses, or potentially the synapses only? This should be made clear upfront.

We have removed panel (a) of Figure 1. The superparamagnetic tunnel junction replaces the entire neuron, which role in our system is to spike at a rate which depends on the received stimulus. The introduction now explains this prominently:

“In the present work, we show that a nanodevice – the superparamagnetic tunnel junction – naturally implements neurons for population coding, and that it can be exploited for designing systems that can compute and learn. The behavior of the nanodevice directly provides a tuning curve and resembles a spiking neuron.”

We propose to implement the synapses with stable magnetic tunnel junctions. These devices are the same technology as the superparamagnetic junctions (but larger), which would allow using one stack of materials for both neurons and synapses. Magnetic tunnel junctions are used as non-volatile memory cells in MRAM, which is a mature and already commercialized technology.

Neurons and synapses here need to be connected by CMOS circuits, in particular to compute the rates of the neurons and write the synaptic weights. We address this point in our responses to your later questions.

We have clarified these points in the text:

“To evaluate the viability of the approach, we designed a full system associating superparamagnetic tunnel junctions as input neurons, CMOS circuits and standard magnetic tunnel junction used as spin torque magneto-resistive memory (ST-MRAM) to store the synaptic weights w_{ij} . These stable junctions could be fabricated using the same magnetic stacks as the superparamagnetic junctions (but a different sizing).”

3) Also, the presented approach is only one of many types of population coding. Since population coding is central to the publication, a clear positioning of the work with respect to other types of population coding should be provided rather than vague/ambiguous statements ("in particular, a neuron inside a population should possess a tuning curve").

We have added in the introduction some background about population coding and why it is attractive:

"A key for the resilience of the brain seems to be redundancy. Measurements of neuronal activity in diverse parts of the brain such as the retina⁶, the midbrain⁷, the motor cortex⁸ or the visual cortex⁹ indicate that they encode and process information by populations of neurons rather than by single neurons. This principle of population coding and its benefits for the brain have been investigated by numerous theoretical works^{10,11} "

Furthermore, in the discussion, we have positioned our work with respect to the various ways to do population coding:

"Our work reproduces the essence of population coding in neuroscience, with some adaptations for implementation with nanoelectronics. In population coding theory, neuronal correlation^{11,45}, the meaning of the time¹¹, as well as decoding techniques⁴⁵ are contentious topics. In our system, these aspects were guided by the properties of the nanodevices and by circuit design principles. The input neurons spike in an uncorrelated fashion, as their noise originates from basic physics. The time is divided into discrete phases, allowing the use of counters and finite state machines in the system. The information is decoding by counting spikes using simple unsigned digital counters."

4) It seems rather unclear how the actual hardware implementation would look. In hardware-based emulations of neurons and synapses, the electronic components necessary to "run" the neuron are of importance, since they determine the compactness and scalability of the proposed solution.

We have designed the full computation system, including both nanodevices and CMOS circuitry. We have added a "Design of the full system" section in the main text. Figure 5 (a) presents a schematic of the architecture and Figure 5 (b) and (c) present the energy and area consumptions of the various parts. More details about the CMOS circuits as well as how they were designed and how the energy consumption was computed are presented in the Methods. Section 5 of the Supplementary Information describes the full data-path of the system. Section 6 of the Supplementary Information gives more details about the energy and area consumption of the system.

How would the firing of the neuron be detected (drop in resistance), how would be the rates computed, represented and communicated, how would the neuron be interfaced to synapses or other neurons?

We have now explained these parts in the main text:

"Junctions switching events are detected by a CMOS circuit comparing the voltage across a junction and the corresponding voltage on a reference resistance (see Methods). Each junction is associated with a digital counter counting the switches. After a stimulus operation phase, the system can compute its

output using Eq. (4) using integer arithmetics. This is done by a digital circuit that we designed and is described in Methods. The synaptic weights w_{ij} are stored in stable magnetic tunnel junctions assumed in a 28 nm technology (see Methods). If the system is in a learning phase, the learning rule is then applied by a digital circuit, also described in Methods, which reprograms some ST-MRAM cells.”

What is the minimum and maximum operational speed (firing rate, spike rate, ...?) of the neuron?

The spike rate of the neurons is determined by the energy barrier of the superparamagnetic tunnel junctions as described by Equation 1. The junctions used for the design of the full system and the energy and area consumption mentioned above were assumed to have an energy barrier of $\Delta E = 6k_B T$, which corresponds to a rate of 1.23 MHz. Faster junctions lead to a better precision for a fixed observation time because more switches are available to compute the rate. Besides, faster junctions are usually of lower diameter and therefore require less current to be controlled.

The derivation of maximum operational speed now appears in the Methods section.

In this respect, computer-based simulations have limited validity since some operations (e.g. matrix inversion) are computationally heavy. Again, it is very useful to outline how the proposed neurons would fit into a computational system.

Matrix inversion was indeed used to compute the weights generating the handwritten letters from the experimental tuning curves. This is indeed computationally heavy and unpractical for applications. In consequence we propose a learning scheme, presented in the gripper study, which does not require matrix inversion. We made this point more explicit in the Methods:

“In Figs. 3 and 4, the weights are obtained by the learning process and no matrix inversion is necessary.”

5) The authors mention multiple times (e.g. in the first sentence of the abstract and in the conclusion) the notion of fault-tolerant information processing and device variability. Therefore, I think it would be a good idea to include representative measurements of the device variability and endurance upfront when the device is discussed (independent of the simulation-based studies), and discuss how the fault-tolerance of the population is impacted in terms independent of the simulation-based gripper study.

We have included this information as suggested:

“As can be observed from Fig. 2a, the junctions are not identical due to the polycrystalline nature of the free ferromagnetic layer (see Methods). This variability affects both the critical current I_c and the energy barrier ΔE , resulting in the width variations of the tuning curves in Fig. 1a, but also in different natural rates that for this set of junctions span between a few Hz and 70 kHz.”

Figure 4 (b) quantitatively studies the effect of the variability on the energy barrier.

Section 2 of the Supplementary Information gives further details about the effect of the variability.

6) Later on, the study can show that a break-down of some neurons or synapses (not shown) or their significantly biased performance (shown) does not hurt the overall performance of the population on a particular task.

In the Supplementary Information (section 3) we have added a section about the effect of the breakdown of neurons on the gripper task. We show that our computing system is resilient to the loss of neurons.

7) In the gripper study, it was not clear to me whether the authors aim at presenting an improved learning algorithm on top of the more hardware-level findings ("The key advantage is ... no need to perform a precise measurement...as required by most learning methods in the literature.").

As pointed out by reviewer 3, this learning algorithm was proposed in other studies (in particular of population coding with CMOS neurons). Other learning algorithms can be proposed, in particular to go to more complex multi-layer architectures. However, we wanted to highlight that an extremely simple learning algorithm can allow computing of non-linear functions, demonstrating the power of population coding.

We have modified the text to make these points clearer:

"For this purpose, we follow an error and trial procedure, similar to the one described in³⁰."

and

"Note that the proposed system is independent of this learning rule and that different algorithms could be used to perform more difficult tasks."

8) Are the results of the simulation study specific for the junction-based neuronal network? It may seem the fact that increasing the size of a basis set improves a signal approximation is rather a generic finding and is intuitive.

The results of the simulation study are specific for the junction based neuronal network because they use a model of superparamagnetic tunnel junction (a Poisson process which escape rates are modified by spin transfer torque) and parameters and variability extracted from experiments. However we would expect other Poisson oscillators to give very qualitatively similar results.

It is true that improving the precision by improving the size of the basis set is generic to population coding (ref 30 of the article). However, we wanted to make the point that superparamagnetic tunnel junctions are promising devices to emulate neurons, and provide elements about how the system should be sized. Furthermore, the fact that increasing the number of junctions improves the basis set and therefore the precision might not be evident for readers from nanodevices fields who are not familiar with population coding.

9) If I understood correctly, the gripper study is solely based on simulations and only using the bell-shaped model of neuronal response that is not the key contribution of the paper (Figure 3), partly with a

feed-in of the STT-related parameters (Figure 4). It would have a much larger expressive value if the neurons would be really implemented in hardware.

We agree that implementing the full computing system in hardware and demonstrating the gripper task experimentally would be great. However this is a longer term project that will require collaborating with architecture design experts and industrial foundry facilities. We believe that the work presented in this paper will trigger many studies on the topic and lead to an actual full hardware realization.

9) I would also recommend to improve the clarity of the figures and of the text. For instance, Figure 1 contains a lot of panels, some of which are actually only the background of the presented work (panels a,b,c,d,e). The first figure should summarize the key contribution in a crisp and concise way. Also, figures without descriptions (panel (a)) are not very informative.

We have removed panel (a), which indeed did not provide information. In the new Figure, panels (a-e) still present neural behavior and population coding which are background elements rather than our work. All of them are properly referenced in the body text.

We propose to keep these panels, as we feel that some of the audience – in particular the community of spin electronics – will not be familiar with the neuroscience background necessary to understand our work. In the many one-to-one presentations of our work we have conducted – and from our own experience of learning about population coding as physicists – we have observed that showing side by side the behavior of neurons and superparamagnetic tunnel junctions is very helpful to attract attention and help understanding. It goes the same way for the schematic presentation of population coding we propose in Fig. 1. Of course, we are open to changing our position

10) The first two paragraphs seem to me too vague at times ("the processor would melt", "the key seems to be redundancy", ...). I think the authors could make a better use of the space by providing the reader with a concise overview of background results and literature, with proper citations and terminology.

We have re-written the first two paragraphs, removed the vague elements, and included more background elements.

11) Also, it might be a good idea to move some parts of the simulations (the gripper study) to Supplementary Information, since dis-proportionally large parts of the text are now occupied by its description.

We have now notably extended the paper with the addition of the full system. As this full system implements the task described in the theoretical gripper study, we have chosen for clarity to keep its description in the body text of the article. However, we are open to change our position if the reviewer thinks that some parts of the study divert the reader from the main message.

Reviewer #2 (Remarks to the Author):

Prof. Grollier's group is well known for their past work on superparamagnetic MTJ's showing results like the ones shown in Fig.1f to 1j. What I believe is new experimentally is the demonstration of biased junctions with shifted tuning curves as shown in Fig.2a along with the proposal for implementing it compactly that is described in the Appendix.

This experimental demonstration plus proposal though interesting is probably not enough by itself to justify publication in Nature Communication. Presumably it is the connection to neural computing that provides this justification. Unfortunately I do not feel convinced by this part of the argument which is purely software based, with no discussion of how it could possibly be done in hardware. Let me elaborate.

The authors propose a hardware implementation of superparamagnetic tunnel junctions for basis functions for a class of problems called 'population coding'. I enjoyed reading the paper, and found the main idea interesting but I feel that there is a strong dissonance between the claims and the actual demonstration.

12) The paper seems to justify its conclusions by drawing parallels to performance of computers for the post-CMOS era, but no actual comparisons are attempted to see how a standard CMOS implementation of their examples would have fared against this highly computer-assisted demonstration.

13) The abstract of the paper claims that strong requirements have until now prevented a demonstration of population coding with nanodevices. It is not clear why a standard nanoscale CMOS implementation of the present examples would have problems generating the non-linearity and stochasticity, especially considering the paper itself makes extensive use of external CMOS hardware that requires non-trivial functions as I point out below.

We thank the reviewer for his/her comments on the manuscript. To address them and give more depth to the manuscript, we have decided to design a full system, associating the superparamagnetic tunnel junctions and the associated CMOS overhead. This work was mostly performed by a new co-author (T. Hirtzlin). This work is described in a new section "Design of the full system" to the main text. We describe there the architecture and CMOS circuits that would be used for a full implementation. More details can also be found in the Methods and in the Supplementary information (Sections 5 and 6). This work allowed us to compare the obtained results in terms of energy and are consumption to pure CMOS implementations:

"It is instructive to compare these results with solutions where neurons would have been implemented with purely CMOS circuits. The reference design of a low-power CMOS spiking neuron of ⁴⁴ provides features similar to our nanodevices (analog input and spiking digital output). If we had employed this design, the neurons would have occupied $128,000 \mu\text{m}^2$, making the neurons the dominant area in the circuit. The energy consumption of the neurons would have been 330 nJ per operation, also the dominant energy consumption of the system. The design of ¹³, by contrast, provides a purely analog (non-spiking) solution. In that case, the neurons would have occupied a more reasonable $1,280 \mu\text{m}^2$ and consumed 200 pJ (assuming the system runs for $10 \mu\text{s}$). However as the neurons have analog outputs, the

processing is more complicated to do in a precise manner, and the design cannot scale without the addition of energy-hungry analog-to-digital converters. Finally, a more conventional solution, using a generic processor and not an application-specific integrated circuit would have naturally used order-of-magnitudes more energy.”

14) Currently, the abstract reads “We demonstrate experimentally that a population of nine junctions can implement a basis set of functions, allowing, for instance, the generation of cursive letters’. I find this statement misleading. If the authors needed 9 MTJs to obtain the main experimental results of the paper (For example, Figure 2c), the demonstration would be an important step towards a novel, post-CMOS computation unit. As it stands, there seems to be an enormous amount of detail that is currently not discussed in the manuscript.

We feel that this legitimate concern has been solved overall by the inclusion of our new design work. The reader can now see exactly what kind of CMOS overhead the junctions require, and why we feel that using them is very advantageous.

Concerning the particular case of the cursive letters generation, we have modified the abstract to avoid the reader being misled:

“We demonstrate experimentally that a population of nine junctions can implement a basis set of functions, providing the data to achieve, for instance, the generation of cursive letters.”

We have also clarified the text to make more obvious the computer-based steps:

“The tuning curves are obtained experimentally. Then this data is used to achieve the transformation to the output function H by performing with a computer a weighted sum through:

$$H(I) = \sum_{i=1}^9 w_i r_i(I), \quad (3)$$

where the optimal weights in Eq. 3 for the desired function H are obtained through matrix inversion on a computer (see Methods). “

Other specific concerns and questions:

15) 1. The starting unit of (Figure 1j) where the Gaussian rate is obtained from superparamagnetic junctions was processed by a computer. Am I correct in noting that this is currently not an experimental quantity that was measured in actual hardware? If so why is it important to achieve the shifting (Fig.2a) in actual hardware?

The oscillations of the resistance of the junction were measured experimentally. Then, a computer was indeed used to compute the rate associated with the measurement at each current value. We have clarified this point in the text:

“Based on measurements such as the ones on Fig. 1(g-i), we extracted the rate of the junction at various current values. The resulting experimental rate versus current curve $r(I)$ is shown in Fig. 1(j).”

In a full hardware implementation, this step will be realized by a small CMOS circuit, described in the section “Design of the full system”:

“Junctions switching events are detected by a CMOS circuit comparing the voltage across a junction and the corresponding voltage on a reference resistance (see Methods). Each junction is associated with a digital counter counting the switches.”

More details have also been added in the Methods and in the Supplementary Information (section 5).

Computing the rate that would have been obtained with a shifted junction, from the rate measured with a non-shifted junction is a complex calculation, therefore shifting the junctions in hardware is a better solution.

16) 2. Given the apparent difficulty in controlling the variations and randomness of the rate-response and heavy-use of external, standard hardware, one could argue that all of the results in the paper could have been obtained from a single MTJ that is shifted externally and then processed (to get the rate), multiplied and summed accordingly.

It is true that using a single MTJ would have been possible. However this implementation would have two drawbacks compared to a multi-MTJ one:

- The computation would become intrinsically sequential and would therefore be longer as the measurement of the rates would have to be done “one junction after the other”. On the contrary, in a population of junctions, the rate measurement can be done in parallel for all junctions.
- The system would be over-sensitive to one single device. Failure of this device would be fatal. Whereas with a population of junctions, the system is resilient to the failure of components, as described in a new section in the supplementary information. Furthermore, in a population of junctions, device variability evens out and has not critical effect on the system whereas a single junction implementation would require perfect control over the parameters of the junction.

We have clarified this point in the text:

“In order to build a population, we need to tune each junction to different ranges of input currents. Having a full population rather than a single superparamagnetic tunnel junction allows for parallel processing of each neuron as well as resilience to the inevitable variability – or even failure – of the devices (see Supplementary Information, sections 2 and 3).”

17) 3. Figure 2a shows the processed response of 9 MTJs as a function of a current. Figure 2c shows six examples of cursive letters that are obtained by this data. Since the hardware currently does not

calculate the weights, the multiplication of the weights with the basis functions, nor the summation, there is a substantial use of standard CMOS technology.

The cursive letters are here to highlight the ability of populations to represent non-linear functions by a simple linear combination. It is true that here the weights have been here obtained by matrix inversion, which is computationally expensive. This is why we propose a learning algorithm which does not require matrix inversion to get the weights, and use it in our simulations of the gripper study.

18) 4. If the authors are envisioning a comprehensive hardware implementation, they should at least provide a sketch for implementing the weights, multiplication and summation of the basis functions.

We have now designed the full computation system, including both nanodevices and CMOS circuitry. We have added a “Design of the full system” section in the main text. Figure 5 (a) presents a schematic of the architecture and Figure 5 (b) presents the energy and area consumptions of the various parts. More details about the CMOS circuits as well as how they were designed and how the energy consumption was computed are presented in the Methods. Section 5 of the Supplementary Information describes the full data-path of the system. Section 6 of the Supplementary Information gives more details about the energy and area consumption of the system.

These results include how to detect the firing of the neuron, how to compute the rates, how to compute the output of the system using the values stored in the synaptic weights, how to implement learning and program the synaptic weights. The roles and place of the different technologies (stochastic junctions, ST-MRAM and CMOS) are clearly identified.

19) In short, it seems to me that the basic information processing concepts discussed here are well-known and could be implemented all in software. The authors seem to be replacing an insignificant fraction of this with biased MTJ's. The rest still needs to be implemented in software. Perhaps I am missing something. If so, perhaps the authors can clearly articulate what their contribution is ?

Although population coding has been broadly studied in neuroscience, hardware implementations of it are very few and using a superparamagnetic tunnel junction as artificial neuron for it is definitely new. We hope that the inclusion of our new sections shows the relevance of the approach. We have also entirely rewritten the introduction of the paper and extended its discussion so that the contribution is clearer.

In summary, because they naturally implement rate coding and exhibit a non-linear tuning curve, superparamagnetic tunnel junctions do replace the most important features of neurons for population coding. In the section “Design of the full system” we show that our system present improvements compared to pure CMOS implementations of neurons, in particular in terms of energy and are consumption.

While traditional full CMOS architectures are probably the best for general purpose computing, we think that our system is competitive for specific applications where energy and area consumption are critical factors, such as smart sensors or wearable devices for example.

Reviewer #3 (Remarks to the Author):

This paper shows nanoscale magnetic tunnel junctions, which exhibits neuron-like Gaussian tuning curves. Authors have used these tuning curves for pattern recognition tasks in software. However, the reviewer has some concerns/comments as below.

We thank the reviewer for his/her careful reading of the manuscript.

20) 1. There are some works on building CMOS circuit based on population coding:

A neuromorphic hardware framework based on population coding. IEEE International Joint Conference on Neural Networks (IJCNN).

A Low Power Trainable Neuromorphic Integrated Circuit that is Tolerant to Device Mismatch. IEEE Transactions on Circuits and Systems I: Regular Papers, 63(2), 211-221.

We have included references to this works in the paper.

"Mimicking population coding has also been proposed in circuits using conventional transistors^{13,14"}

21) 2. Ref [8] was not formatted correctly.

We reformatted ref 8 appropriately.

22) 3. Sentence "Measurements of neuronal activity indicate...processed by population...rather than single.." is not entirely correct. This is just one kind of information encoding in the brain.

We have clarified the text:

"Measurements of neuronal activity indicate that in the brain, one method used is to encode and process information by populations of neurons rather than by single neurons³."

23) 4. Figure 1 part (i) $I = -10\mu A$ but in the figure, it is $=0$. Needs to change.

We have modified the figure 1 accordingly.

24) 5.Sentence "random resistive switches follow a Poisson process.." is it modeled or is it an assumption?

This has been modeled. In previous work (Locatelli et al., Phys. Rev. Applied, 2014) we show histograms of the dwell times of parallel and antiparallel states, exhibiting a Poisson distribution. This model has also been used in various works (Rippard et al., Phy. Rev. B, 2011 ; Li and Zhang, Phys. Rev. B, 2004).

25) 6.Sentence "shifting the tuning curves can also be achieved by applying..." check the below paper, where a similar idea is proposed...consider citing it.

A neuromorphic hardware framework based on population coding. IEEE International Joint Conference on Neural Networks (IJCNN).

We have cited this paper at the suggested place:

"This method has been used in other works of hardware implementation of population coding²⁹."

26) 7.Authors have shown the architecture for single input case. How the multiple inputs will be combined in this architecture. Please elaborate.

In Figure 4 (a) we show results for a multi-inputs computation. The methods explicit how to perform this computation:

"The two input populations R and φ are concatenated into a single population. Its number of junctions is the sum of the number of junctions in each population $N_{IN} = N_R + N_\varphi$. Two weights matrices (W_x and W_y) connect the input (R , φ) to the output junctions (x , y). The weights matrices W_x and W_y have the dimensions $N_x \times N_{IN}$ and $N_y \times N_{IN}$. Where N_x (N_y) is the number of junction encoding x (y). Learning of the weights is implemented as described previously."

We have also added a schematic of the required architecture in the Supplementary Information (section 4).

27) 8.The output of the nanoscale devices are spikes..how is it converted into rates using circuits. This needs to be elaborated.

We have designed a circuit able to detect the spikes and convert them into rates.

"Junctions switching events are detected by a CMOS circuit comparing the voltage across a junction and the corresponding voltage on a reference resistance (see Methods). Each junction is associated with a digital counter counting the switches."

Details about how this circuit was designed can be found in the Methods. Information about the energy and are consumption of this circuit can be found in the main text and the Supplementary Information (section 6).

28) 9. The system seems to be a three-layer neural network. It would be helpful for readers if a system level diagram is provided where nanoscale devices could be a neuron. Also, how the weights would be trained between the layers, should be elaborated in the fig.

Our system is actually a two-layer neural network. The input and output layers of neurons are connected by synapses (that we propose to implement with ST-MRAM). Figure 5(a) and the Supplementary Information (section 5) explain how the training of the weights is implemented.

However, this computing unit can be combined with others in order to form more complex networks. For instance, the computation $(\text{sine})^2$ (Fig. 4a, label “Series”) was obtained by a three-layer network. We have added a schematic of this in the Supplementary Information (section 4).

29) 10. Weight modification rule is similar to a previous work as shown below: An Online Learning Algorithm for Neuromorphic Hardware Accelerators...arXiv:1505.02495

We have now referenced this work:

“For this purpose, we follow an error and trial procedure, similar to the one described in³⁰”.

30) 11. The main concern is how the multi-input system will be built from the tuning curves. The reviewer has some concerns over its practical implementation. Please justify.

In our architecture, each input will require its own population of junctions (i.e. its own basis set of tuning curves). These input populations are then connected to the single or multiple output populations, as described in the Methods and in Supplementary Information (section 4).

31) 12. Authors should specify the advantages of the nanoscale devices over existing devices such as CMOS implementation or others such as speed and power efficiency.

Figure 5(b-c) gives energy and area consumption estimates of our system (more details can be found in the Supplementary Information, section 6). We use these, as well as other considerations to show the advantages of our system compared to CMOS implementations:

“It is instructive to compare these results with solutions where neurons would have been implemented with purely CMOS circuits. The reference design of a low-power CMOS spiking neuron of⁴⁴ provides features similar to our nanodevices (analog input and spiking digital output). If we had employed this design, the neurons would have occupied $128,000 \mu\text{m}^2$, making the neurons the dominant area in the circuit. The energy consumption of the neurons would have been 330 nJ per operation, also the dominant

energy consumption of the system. The design of ¹³, by contrast, provides a purely analog (non-spiking) solution. In that case, the neurons would have occupied a more reasonable 1,280 μm^2 and consumed 200 pJ (assuming the system runs for 10 μs). However as the neurons have analog outputs, the processing is more complicated to do in a precise manner, and the design cannot scale without the addition of energy-hungry analog-to-digital converters. Finally, a more conventional solution, using a generic processor and not an application-specific integrated circuit would have naturally used order-of-magnitudes more energy.”

List of changes

Article

- We added a new section “Design of the full system” where the full architecture of the computing unit, including both nanodevices and CMOS circuits, is described. This section, in addition to the design itself, includes circuit simulations estimating the energy and area consumption of the system. These results allowed us to add a comparison of our system to pure CMOS implementations, as asked by all Reviewers. A new section in the Methods, describing how these simulations were conducted, was added.
- A new figure (Figure 5) was added, describing the architecture of the system, as well as its energy and area consumption.
- We fully rewrote the discussion to clarify the positioning of our work regarding population coding and neural nets, as asked by Reviewer 1.
- We fully re-wrote the introduction to address the concerns of Reviewer 1: we have highlighted existing literature.
- We made clearer how the tuning curve in Figure 1 is obtained and how computations in Figure 2 are made, to address the concerns of Reviewer 2. The latter point was also clarified in the Methods.
- We modified Figure 1, as asked by Reviewer 1, and its legend accordingly.
- We moved the explanation of how to decode a population from the Methods to the main text, for clarity.
- We modified the author list to include T. Hirtzlin who participated to the design of the full architecture.
- We made small changes to the abstract to account for the new sections in the article.
- We updated references to include those recommended by the Reviewers.

Supplementary information

- We added a new section “Schematic of the systems allowing more complex transformations” to clarify how to implement multi-inputs transformations and three-layers networks as asked by Reviewer 3.

- We added a new section “Data path of the full system” which describes in detail the full architecture of the system, thus addressing a concern raised by all reviewers.
- We added a new section “Area and energy efficiency of variations of the full system” which estimates the consumption of our system in various cases. This allows to benchmark our system, as asked by all reviewers.

Reviewers' Comments:

Reviewer #1:

Remarks to the Author:

The authors have responded in detail to the reviewers' questions and suggestions, and have updated their manuscript extensively.

Most importantly, the introduction and discussion have been streamlined and updated to properly position the work in the context of other related research. Also, the newly added system-level design enables the reader to assess how the proposed technique could be actually realized. The simulation results on power/energy consumption are encouraging (although, I would try to avoid putting "0" consumption; it just shows the plot scaling is not correct). The supplementary material contains additional information on neuron variability and loss and other information.

After editorial-level correction of language, I think the manuscript is suitable for publication in Nature Communications.

Reviewer #2:

Remarks to the Author:

The authors have done considerable extra work and addressed some of my earlier concerns. In particular, they are now showing precise details of the hardware CMOS periphery one can presumably use for a complete demonstration in Fig. 5.

- The most interesting addition to the paper Fig. 5 now shows a block-diagram view of sophisticated signal processing units that can achieve "population encoding" with nanodevices.

The authors make comparisons with two approaches compared to their own proposal: One with a digital spiking output, the other with an analog output. The digital neuron naturally consumes much more area (while probably being orders of magnitude more reliable than analog designs, the authors did not mention this) but the analog (or mixed-signal) neuron estimations using Ref. 13 seem to produce area and power calculations that are seemingly better than (or comparable to) their own projections. They add that an analog design cannot scale without A2D converters — but isn't their neuron analog too? Aren't they using heavy post-processing as well? Even the basic building block (firing rate) goes through post-processing. It is really not clear to me how this design is more scalable than a standard mixed-signal ASIC. In light of these, can they justify the statement in their abstract:

"These strong requirements have prevented a demonstration of population coding with nanodevices"?

Did they really propose a solution that could not otherwise be done in a practical CMOS-method in this paper? The added section and the abstract seem to disagree with each other.

- The authors confirm that in principle a single MTJ data could have been used to obtain all the results in the paper but note that this would produce sequential outputs and make the system error-prone. This would have been more convincing in the absence of heavy external signal processing – with all kinds of software processing they used here – once again it is not clear to me one could not have parallelized or processed the data from a SINGLE MTJ. But I may be missing something.

- The authors conclude by saying "hardware implementations of [population encoding] are very few and using a superparamagnetic tunnel junction as artificial neuron for it is definitely new".

This might be very true but the abstract for this paper (and justification for the very broad audience Nature Communications provides for that matter) clearly is not phrased in this way.

If the paper were articulated in this way, putting the relatively comparable yet immediately available CMOS options front and center, I would not have any objections.

Reviewer #3:

Remarks to the Author:

1. Authors have shown the architecture for single input case. How the multiple inputs will be combined (with different weights, followed by weighted sum) in this architecture. Please elaborate.
2. The output of the nanoscale devices are spikes..how is it converted into rates using circuits. This needs to be elaborated.
3. The main concern is how the multi-input system will be built from the tuning curves. The reviewer has some concerns over its practical implementation. Please justify
4. line no 329-335, not sure how the authors have come up with the area and energy numbers. In the referred paper [13]. The area of the chip is 1mm^2 , which includes 456 CMOS neurons including the on-chip learning algorithms and digital weights.
5. Also the statement "However as the neurons have analog outputs, the processing is more complicated to do in a precise manner, and the design cannot scale without the addition of energy-hungry analog-to-digital converters." is not clear. It seems the whole idea of making an analog learning chip is to avoid ADCs. Ofcourse the computation is not precise and that might be the reason to have on-chip learning rule to compensate the effect of any nonlinear behaviour.
6. In fig5, It is clearly shown that tunnel junction has a very small imprint (or small part of the overall system) in terms of area, which shows you are not gaining much as compared to the complete CMOS solution because the CMOS based neuron model in [13] uses only 5 transistors, which will not be significantly bigger in low process node.
7. Also, tunnel junction are modelled as a spiking neurons, but the complete system is rate based. Spikes are not used for computation. What is the motivation to have spiking neurons?

We would like to thank the anonymous reviewers for their careful reading of our manuscript, and their comments that allowed us to improve the manuscript significantly. Notably, we have now included a much more comprehensive comparison with purely CMOS solutions, which strengthens our conclusions and makes this manuscript a significant addition to the field of emerging computing devices.

Below are the responses to Referee's questions and comments. We hope that this new version will have addressed all concerns raised in their remarks.

Reviewers' comments:

Reviewer #1 (Remarks to the Author):

The authors have responded in detail to the reviewers' questions and suggestions, and have updated their manuscript extensively.

Most importantly, the introduction and discussion have been streamlined and updated to properly position the work in the context of other related research. Also, the newly added system-level design enables the reader to assess how the proposed technique could be actually realized. The simulation results on power/energy consumption are encouraging (although, I would try to avoid putting "0" consumption; it just shows the plot scaling is not correct). The supplementary material contains additional information on neuron variability and loss and other information.

After editorial-level correction of language, I think the manuscript is suitable for publication in Nature Communications.

We thank the reviewer for his/her evaluation.

The graph has been updated so that "0" consumption does not appear any more.

Reviewer #2 (Remarks to the Author):

The authors have done considerable extra work and addressed some of my earlier concerns. In particular, they are now showing precise details of the hardware CMOS periphery one can presumably use for a complete demonstration in Fig. 5.

The most interesting addition to the paper Fig. 5 now shows a block-diagram view of sophisticated signal processing units that can achieve "population encoding" with nanodevices.

We thank the reviewer for his/her comments.

The authors make comparisons with two approaches compared to their own proposal: One with a digital spiking output, the other with an analog output. The digital neuron naturally consumes much

more area (while probably being orders of magnitude more reliable than analog designs, the authors did not mention this) but the analog (or mixed-signal) neuron estimations using Ref. 13 seem to produce area and power calculations that are seemingly better than (or comparable to) their own projections. They add that an analog design cannot scale without A2D converters — but isn't their neuron analog too? Aren't they using heavy post-processing as well? Even the basic building block (firing rate) goes through post-processing. It is really not clear to me how this design is more scalable than a standard mixed-signal ASIC.

A core element of our concept, which was explained in an implicit way and is now explained in detail, is that our nanodevice-based neurons provide a form of stochastic analog to digital conversion, without the need of an explicit analog to digital conversion.

To improve our paper, we have entirely overhauled the comparison of our approach with alternative approaches. We have added a Supplementary Information section 7, which provides more detailed comparison between our approach and four purely CMOS possible implementations. The body text of the paper is now clearer and more comprehensive:

"It is instructive to compare these results with solutions where neurons would have been implemented with purely CMOS circuits. A detailed comparison to four different approaches is presented in Supplementary information, section 7. A natural idea is to replace our junctions and their read circuitry by low-power CMOS spiking neurons, such as those of⁴⁵, which provides features similar to our nanodevices (analog input and spiking digital output). This strategy works but has high area requirements ($>1\text{mm}^2$), and would consume more than 330 nJ per operation. Alternative options rely on analog computation, for example exploiting neurons such as¹³. Such solutions require the use of an explicit Analog to Digital conversion (ADC), which actually becomes the dominant source of area and energy consumption. Even extremely energy efficient ADCs⁴⁶ require a total of 20 nJ/conversion and an area of 0.2mm^2 . Finally, a more conventional solution, using a generic processor and not an application-specific integrated circuit would have naturally used order-of-magnitudes more energy.

The low energy consumption of our system arises from a combination of three major factors. The superparamagnetic junctions consume a negligible energy (150 pJ), and allow avoiding the ADC bottleneck present in other approaches by implementing a form of stochastic analog to digital conversion in a particularly efficient manner. The use of a stochastic approach and of integer arithmetic in the CMOS part of the circuit is particularly appealing in terms of energy consumption. Finally, associating both CMOS and spintronic technology on-chip limits data transfer-related energy consumption. "

In light of these, can they justify the statement in their abstract: "These strong requirements have prevented a demonstration of population coding with nanodevices"? Did they really propose a solution that could not otherwise be done in a practical CMOS-method in this paper? The added section and the abstract seem to disagree with each other.

We have clarified the abstract so that it matches exactly the content of the paper. In particular, the sentence that concerned the reviewer and the next one have been replaced by:

"These features can be implemented with CMOS technology, but the corresponding circuits tend to have high area or energy requirements. Here, we show that nanoscale magnetic tunnel junctions can instead be assembled to meet these requirements."

Our approach can indeed be implemented by CMOS, and the different CMOS-based roads are now well listed in the new Supplementary Information section 7. Our point is that relying on magnetic devices can allow much smaller area and better energy efficiency than purely CMOS approaches.

The authors confirm that in principle a single MTJ data could have been used to obtain all the results in the paper but note that this would produce sequential outputs and make the system error-prone. This would have been more convincing in the absence of heavy external signal processing – with all kinds of software processing they used here – once again it is not clear to me one could not have parallelized or processed the data from a SINGLE MTJ. But I may be missing something.

To clarify our answer, we have now added a full paragraph in the paper about the benefits and drawbacks of the multiple MTJ and single MTJ approaches:

“It is also a possibility to design the system using a single superparamagnetic junction, and to implement the population response through time multiplexing. This approach would allow avoiding the effects of device variability. However, it would also increase conversion time by the number of input neurons, giving a very low bandwidth to the system. As the superparamagnetic junctions have low area and the system features a natural resilience to device variability, we propose to physically implement the population with an actual population of junctions.”

The authors conclude by saying “hardware implementations of [population encoding] are very few and using a superparamagnetic tunnel junction as artificial neuron for it is definitely new”.

This might be very true but the abstract for this paper (and justification for the very broad audience Nature Communications provides for that matter) clearly is not phrased in this way. If the paper were articulated in this way, putting the relatively comparable yet immediately available CMOS options front and center, I would not have any objections.

We hope that with the reformulation of the abstract and of the discussion, and our now extensive comparison with CMOS options, the original contribution of the paper is now very clear.

Reviewer #3 (Remarks to the Author):

1. Authors have shown the architecture for single input case. How the multiple inputs will be combined (with different weights, followed by weighted sum) in this architecture. Please elaborate.
3. The main concern is how the multi-input system will be built from the tuning curves. The reviewer has some concerns over its practical implementation. Please justify

We thank the reviewer for his/her comments. In fact, several options are possible to manage multiple inputs. It is possible to combine the inputs before presenting them to superparamagnetic tunnel junctions, consistently with the approach recently presented in Ref. 43. This method has the

advantage to limit the total number of junctions and associated read circuitry. However, since this circuit has a small area, we investigated another approach: each input is associated with its own population of superparamagnetic tunnel junctions. The rates originating from different populations can then be sent as inputs to a single neural network, trained to perform operations dependent of the different inputs. This general principle is illustrated in Supplementary Information, section 4, and a possibility of implementation within our hybrid CMOS/nano architecture is now presented in the new version of Supplementary Information, section 8.

As an example, we trained a 2-input system to transform coordinates from polar to Cartesian. The result is shown in Fig 4(a) (bin “2 Inputs”).

For clarification, we have now included a full discussion of two-inputs systems:

“The system can also be adapted for learning and performing tasks involving several inputs. A possible solution to process multiple inputs with a population is to combine them in a single input that can then be presented to the superparamagnetic tunnel junctions, consistently with the approach recently presented in ⁴³. Here we propose a different approach where each input is sent to a different input population, and the rates originating from these separate populations are combined into a single neural network (see Methods and Supplementary Information, section 4). In this way, by using several populations as inputs and outputs, multi-input multi-output computations, and therefore transformations in several dimensions can be learned. In particular, we used this approach to learn the conversion of coordinates from polar to Cartesian system. The results corresponding to this task are labelled “2 inputs’ in Fig. 4(a)”

Also, in the description of the hybrid CMOS/nano system, we have added:

“As presented the circuit features a single input. It may be extended to several inputs, following the principle presented in Supplementary Information, section 8.”

2. The output of the nanoscale devices are spikes. how is it converted into rates using circuits. This needs to be elaborated.

This was indeed only briefly explained in the methods section. We have now extended the methods section, the corresponding description in the body text, and added the corresponding circuit in Supplementary Information.

The overhauled methods section reads:

“The superparamagnetic junctions were modeled based on the previous Methods section, assuming $d=11$ nm diameter, a size that has been demonstrated experimentally ⁵⁴. The energy consumption for the detection of the spikes was based on Cadence Spectre simulation of a simple circuit, presented in Supplementary Information 5, Figure S8bis, and based on the stimulus value corresponding to the highest energy consumption. The stimulus is applied to reference resistors whose resistance is intermediate between the parallel and anti-parallel state resistance of the superparamagnetic tunnel junctions, as well on the superparamagnetic tunnel junction. At each clock cycle, the voltage at the junction and at the reference resistor is compared by a low power CMOS comparator (Fig. S8bis). Simple logic comparing the result of the comparison to the same result at the previous clock cycle allows detecting the junction switching events, which are counted by an eight-bit digital counter. (Each junction is associated with one counter). “

In the body text we have also rewritten the corresponding description:

“Junctions switching events are detected by a CMOS circuit to determine the rates r_i . It consists in a synchronous low power comparator, which compares the voltage across a junction and the corresponding voltage on a reference resistance (see Methods), as well as edge detection logic. Each junction is associated with a digital counter counting the switches.”

The corresponding circuit is now presented explicitly in Supplementary Information 5, Figure S8bis. Supplementary Information 5 also highlights the limitation of our approach:

“This design is not able to detect multiple switching occurring during a single clock cycle. We saw on system-level simulations that this particularity has no impact on the full application.”

4. line no 329-335, not sure how the authors have come up with the area and energy numbers. In the referred paper [13]. The area of the chip is 1mm^2 , which includes 456 CMOS neurons including the on-chip learning algorithms and digital weights.

These numbers are based on Table I in Ref. 13. Surprisingly, this table is only present in the published version of the article. The preprint published on *arxiv.org*, which comes up naturally when looking for the reference in Google Scholar, does not have Table I.

5. Also the statement "However as the neurons have analog outputs, the processing is more complicated to do in a precise manner, and the design cannot scale without the addition of energy-hungry analog-to-digital converters." is not clear. It seems the whole idea of making an analog learning chip is to avoid ADCs. Of course the computation is not precise and that might be the reason to have on-chip learning rule to compensate the effect of any nonlinear behaviour.

6. In fig5, It is clearly shown that tunnel junction has a very small imprint (or small part of the overall system) in terms of area, which shows you are not gaining much as compared to the complete CMOS solution because the CMOS based neuron model in [13] uses only 5 transistors, which will not be significantly bigger in low process node.

Due to this comment, and similar comments of reviewer #2, we have added a new Supplementary Information section 7, which extends this discussion.

We now compare different CMOS-only implementations. Our major point is that purely analog implementation will require an analog to digital conversion (which can be performed at different levels). In such a low power system, the analog to digital conversion will typically be the dominant source of area and energy consumption. In the new supplementary information section 7, we highlight that our junctions provide a form of stochastic analog to digital conversion, which appears to be a lot more energy efficient. We also highlight that providing the same feature than our junctions in CMOS takes more area and energy.

We have also entirely overhauled the discussion about comparison with CMOS options in the body text:

“It is instructive to compare these results with solutions where neurons would have been implemented with purely CMOS circuits. A detailed comparison to four different approaches is presented in Supplementary information, section 7. A natural idea is to replace our junctions and their read circuitry by low-power CMOS spiking neurons, such as those of⁴⁵, which provides features similar to our nanodevices (analog input and spiking digital output). This strategy works but has high area

requirements ($>1\text{mm}^2$), and would consume more than 330 nJ per operation. Alternative options rely on analog computation, for example exploiting neurons such as¹³. Such solutions require the use of an explicit Analog to Digital conversion (ADC), which actually becomes the dominant source of area and energy consumption. Even extremely energy efficient ADCs⁴⁶ require a total of 20 nJ/conversion and an area of 0.2mm^2 . Finally, a more conventional solution, using a generic processor and not an application-specific integrated circuit would have naturally used order-of-magnitudes more energy.

The low energy consumption of our system arises from a combination of three major factors. The superparamagnetic junctions consume a negligible energy (150 pJ), and allow avoiding the ADC bottleneck present in other approaches by implementing a form of stochastic analog to digital conversion in a particularly efficient manner. The use of a stochastic approach and of integer arithmetic in the CMOS part of the circuit is particularly appealing in terms of energy consumption. Finally, associating both CMOS and spintronic technology on-chip limits data transfer-related energy consumption. “

7. Also, tunnel junction are modelled as a spiking neurons, but the complete system is rate based. Spikes are not used for computation. What is the motivation to have spiking neurons?

We think that our overhauled discussion section largely clarifies these two question. Our device not only implements a neuron, but also allows easily converting an analog current to digital spikes, acting as a form of stochastic analog to digital converter.

For clarity, we have also overhauled the discussion of the spiking nature of our junctions:

“It is also important to note that in our system, the junctions act as a form of spiking neurons that employ rate coding, similarly to several population coding theories^{10,11}. The spiking nature of the neurons offers considerable benefits to the full system: it naturally transforms an analog signal into easy-to-process digital signals. The stochastic nature of the neurons is one of the keys of the energy efficiency and of the robustness of the system. It also gives the possibility for the system to provide an approximate or precise answer depending on the time and energy budget, similarly to stochastic computing^{43,47}. The rest of the system is rate based, which allows learning tasks in a straightforward manner. Another possibility would have been to perform the entire operation in the spiking domain, as is common in the neuromorphic engineering community⁴⁹⁻⁵¹. However, learning in the spiking regime remains a difficult problem today⁴⁸ and involves more advanced concepts and overheads⁵¹. Therefore, our system is designed to takes benefits from both the spiking and the rate-coding approaches.”

Reviewers' Comments:

Reviewer #2:

Remarks to the Author:

I applaud the authors for a major overhaul of the paper and doing their best efforts for a proper comparison with CMOS technology. I think the paper has a clearer perspective over its contributions with its key advantages laid out in detail. Arguably one disadvantage is that CMOS technology is "here and now" that can be scaled on demand, while any emerging technology has to be thoroughly justified to compete with CMOS and perhaps this aspect could have been spelled out explicitly to make sure an unfamiliar audience understands the facts. Nevertheless, I am leaving this to the judgment of the authors and recommend publication.

We thank all the anonymous reviewers for the time they dedicated careful reading of our manuscript, and their comments that allowed us to improve the manuscript significantly.